# GSE: GROUP-WISE SPARSE AND EXPLAINABLE ADVERSARIAL ATTACKS

**Shpresim Sadiku**[1,2]**, Moritz Wagner**[1,2] **& Sebastian Pokutta**[1,2]
[1]Department for AI in Society, Science, and Technology, Zuse Institute Berlin, Germany
[2]Institute of Mathematics, Technische Universität Berlin, Germany
`{sadiku,wagner,pokutta}@zib.de`

## ABSTRACT

Sparse adversarial attacks fool deep neural networks (DNNs) through minimal pixel perturbations, often regularized by the $\ell_0$ norm. Recent efforts have replaced this norm with a structural sparsity regularizer, such as the nuclear group norm, to craft group-wise sparse adversarial attacks. The resulting perturbations are thus explainable and hold significant practical relevance, shedding light on an even greater vulnerability of DNNs. However, crafting such attacks poses an optimization challenge, as it involves computing norms for groups of pixels within a non-convex objective. We address this by presenting a two-phase algorithm that generates group-wise sparse attacks within semantically meaningful areas of an image. Initially, we optimize a quasinorm adversarial loss using the $1/2-$quasinorm proximal operator tailored for non-convex programming. Subsequently, the algorithm transitions to a projected Nesterov's accelerated gradient descent with $2-$norm regularization applied to perturbation magnitudes. Rigorous evaluations on CIFAR-10 and ImageNet datasets demonstrate a remarkable increase in group-wise sparsity, e.g., $50.9\%$ on CIFAR-10 and $38.4\%$ on ImageNet (average case, targeted attack). This performance improvement is accompanied by significantly faster computation times, improved explainability, and a $100\%$ attack success rate.

## 1 INTRODUCTION

Deep neural networks (DNNs) are susceptible to adversarial attacks, where input perturbations deceive the network into producing incorrect predictions (Carlini & Wagner, 2017; Athalye et al., 2018; Zhou et al., 2020; Zhang et al., 2020). These attacks pose serious security risks in real-world systems and raise questions about the robustness of neural classifiers (Stutz et al., 2019). Investigating adversarial attacks is crucial for diagnosing and strengthening DNN vulnerabilities, especially in areas where such attacks have proven effective, including image classification (Chen et al., 2020; Li et al., 2020), image captioning (Xu et al., 2019b), image retrieval (Bai et al., 2020; Feng et al., 2020), question answering (Liu et al., 2020a), autonomous driving (Liu et al., 2019), automatic checkout (Liu et al., 2020b), face recognition (Dong et al., 2019), face detection (Li et al., 2019), etc.

While many methods for crafting adversarial examples focus on $\ell_p$ neighbourhoods with $p \geq 1$, recent research has explored the intriguing case of $p = 0$, leading to sparse adversarial attacks. The prevailing approaches for generating sparse adversarial attacks involve solving $\ell_0-$formulated problems, employing greedy single-pixel selection (Su et al., 2019), local search techniques (Narodytska & Kasiviswanathan, 2016), utilizing evolutionary algorithms (Croce & Hein, 2019), or relaxing $\ell_0$ via the $\ell_1$ ball and applying various algorithms to handle these structures (Carlini & Wagner, 2017; Modas et al., 2019). However, most of these methods only minimize the number of modified pixels and do not constrain the location and the magnitude of the changed pixels. The perturbed pixels can thus exhibit substantial variations in intensity or colour compared to their surroundings, rendering them easily visible (Su et al., 2019). This has motivated the necessity of imposing structure to sparse adversarial attacks by generating group-wise sparse perturbations that are targeted to the main objective in the image (Xu et al., 2019a; Zhu et al., 2021; Imtiaz et al., 2022; Kazemi et al., 2023). Fig. 1 illustrates successful group-wise sparse adversarial examples generated by our proposed algorithm (GSE). This way the generated perturbations are also explainable, i.e., they perturb semantically meaningful pixels in the images, thus enhancing human interpretability.

Unlike traditional attacks that appear as noise to humans but as features to DNNs (Ilyas et al., 2019), this approach bridges the gap between human perception and machine interpretation.

**Contributions.** ① We devise a two-phase approach for creating group-wise sparse adversarial attacks. First, we employ non-convex regularization to select pixels for perturbation, leveraging a combination of the proximal gradient method from (Li & Lin, 2015) and a novel proximity-based pixel selection technique. Then, we apply Nesterov's accelerated gradient descent (NAG) on the chosen pixel coordinates to complete the attack. ② Our GSE (**G**roup-wise **S**parse and **E**xplainable) attacks outperform state-of-the-art (SOTA) methods on CIFAR-10 and ImageNet datasets, requiring significantly fewer perturbations. We achieve a $50.9\%$ increase in group-wise sparsity on CIFAR-10 and a $38.4\%$ increase on ImageNet (average case, targeted attack), all while maintaining a $100\%$ attack success rate. ③ Through a quantitative assessment of the alignment between perturbations and salient image regions, we underscore the value of group-

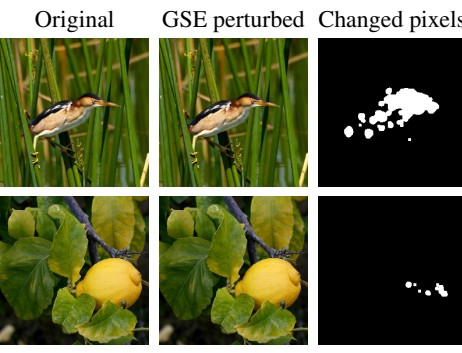

Figure 1: Adversarial attacks generated by our algorithm. The top row depicts a targeted attack of target label "water bottle", and the bottom row depicts an untargeted attack.

wise sparse perturbations for explainability analysis. Specifically, our GSE attacks provide superior explainability over SOTA techniques in both group-wise sparse and traditional sparse attack domains.

## 1.1 RELATED WORK

Recent works have introduced methods for generating group-wise sparse and explainable adversarial attacks. StrAttack (Xu et al., 2019a) represents a structured sparse and explainable adversarial attack, relying on the alternating direction method of multipliers (ADMM). It enforces group-wise sparsity through a dynamic sliding mask designed to extract spatial structures from the image. Similarly, SAPF (Fan et al., 2020) leverages $\ell_p-$Box ADMM for integer programming to jointly optimize binary selection factors, continuous perturbation magnitudes of pixels, and image sub-regions. FWnucl-group (Kazemi et al., 2023), abbreviated as FWnucl, is another structured sparse adversarial attack. It quantifies the proximity of perturbations to benign images by utilizing the nuclear group norm, capitalizing on the convexity of nuclear group norm balls through the application of a Frank-Wolfe (FW) optimization scheme (Frank & Wolfe, 1956). Homotopy-Attack (Zhu et al., 2021) is a sparse adversarial attack that operates based on the non-monotone accelerated proximal gradient algorithm (nmAPG) (Li & Lin, 2015). It can incorporate group-wise sparsity by segmenting the image pixels using the SLIC superpixel algorithm and applying regularization using the resulting $2, 0-$"norm". We empirically compare our approach to the aforementioned attacks.

New research combines adversarial examples and model explanations, merging key concepts from both areas. On datasets like MNIST (LeCun et al., 1998), (Ignatiev et al., 2019) illustrates a hitting set duality between adversarial examples and model explanations, while (Xu et al., 2019a) shows the correspondence of attack perturbations with discriminative image features. Our attack strategy perturbs regions neighbouring already perturbed pixels, resulting in a shift of the most susceptible pixels of an image due to our initial sparse perturbations. We conduct empirical analysis to examine the overlap between our attacks and salient image regions. To highlight the crucial role of group-wise sparse perturbations in enhancing explainability, we also incorporate SOTA sparse adversarial attacks (Modas et al., 2019; Croce & Hein, 2019) into our experiments.

## 2 ADVERSARIAL ATTACK FORMULATION

Let $\mathcal{X} = [I_{\min}, I_{\max}]^{M \times N \times C}$ represent feasible images, where $M$ and $N$ are the image dimensions (height and width) and $C$ is the number of color channels. Let $x \in \mathcal{X}$ be a benign image with label $l \in \mathbb{N}$, and $t \in \mathbb{N}$ be a target label, where $t \neq l$. Additionally, let $\mathcal{L} : \mathcal{X} \times \mathbb{N} \to \mathbb{R}$ be a classification loss function, such as the cross-entropy loss, tailored for a given classifier $\mathcal{C}$. We introduce a distortion

---

**Algorithm 1** Forward-Backward Splitting Attack

---

**Require:** Image $\boldsymbol{x} \in \mathcal{X}$, target label $t$, loss function $\mathcal{L}$, sparsity parameter $\lambda > 0$, step sizes $\alpha_k$, number of iterations $K$.
1: Initialize $\boldsymbol{w}^{(0)} \leftarrow \mathbf{0}$.
2: **for** $k \leftarrow 0, ..., K - 1$ **do**
3:     $\boldsymbol{w}^{(k+1)} \leftarrow \text{prox}_{\alpha_k \lambda \| \cdot \|_p^p} \left( \boldsymbol{w}^{(k)} - \alpha_k \nabla_{\boldsymbol{w}^{(k)}} \mathcal{L}(\boldsymbol{x} + \boldsymbol{w}^{(k)}, t) \right)$         {Definition in Eq. (3).}
4: **end for**
5: **return** $\hat{\boldsymbol{w}} = \boldsymbol{w}^{(K)}$

---

function $\mathcal{D} : \mathbb{R}^{M \times N \times C} \rightarrow \mathbb{R}_{\geq 0}$. The goal of a *targeted adversarial attack* is to find an image $\boldsymbol{x}_{\text{adv}}$ to which $\mathcal{C}$ assigns the target label $t$ and that is in close proximity to $\boldsymbol{x}$ according to the function $\mathcal{D}$. In summary, formulating a targeted adversarial example for the input $\boldsymbol{x}$ can be framed as

$$\min_{\boldsymbol{w} \in \mathbb{R}^{M \times N \times C}} \mathcal{L}(\boldsymbol{x} + \boldsymbol{w}, t) + \lambda \mathcal{D}(\boldsymbol{w}), \tag{1}$$

where $\lambda > 0$ is a regularization parameter. The *untargeted adversarial attack* formulation can be found in Appendix A. Note that Eq. (1) is a general framework, and a specific distortion function $\mathcal{D}$ must be defined to generate adversarial examples in practice.

## 2.1    $1/2-$QUASINORM REGULARIZATION

Sparse adversarial attacks can be generated using quasinorm-regularized methods (Wang et al., 2021), where the distortion function in Eq. (1) is set to $\mathcal{D}(\cdot) := \| \cdot \|_p^p$, leading to the optimization problem

$$\min_{\boldsymbol{w} \in \mathbb{R}^{M \times N \times C}} \mathcal{L}(\boldsymbol{x} + \boldsymbol{w}, t) + \lambda \| \boldsymbol{w} \|_p^p, \tag{2}$$

for $0 < p < 1$. Here $\| \boldsymbol{w} \|_p = \left( \sum_i |w_i|^p \right)^{1/p}$ is only a quasinorm since subadditivity is not satisfied for $p < 1$. Once Eq. (2) is solved, yielding $\hat{\boldsymbol{w}} \in \arg\min_{\boldsymbol{w} \in \mathbb{R}^{M \times N \times C}} \mathcal{L}(\boldsymbol{x} + \boldsymbol{w}, t) + \lambda \| \boldsymbol{w} \|_p^p$, the adversarial example is given by $\boldsymbol{x}_{\text{adv}} = \text{clip}_{\mathcal{X}}(\boldsymbol{x} + \hat{\boldsymbol{w}})$. To obtain $\hat{\boldsymbol{w}}$, we employ the forward-backward splitting algorithm (Li & Lin, 2015), as detailed in Algorithm 1. In their work, (Cao et al., 2013) derive a closed-form solution for the proximal operator of $\| \cdot \|_p^p$

$$\text{prox}_{\lambda \| \cdot \|_p^p}(\boldsymbol{w}) := \arg\min_{\mathbf{y} \in \mathbb{R}^{M \times N \times C}} \frac{1}{2\lambda} \| \mathbf{y} - \boldsymbol{w} \|_2^2 + \| \mathbf{y} \|_p^p, \tag{3}$$

for $p = \frac{1}{2}$. Given that $\| \cdot \|_p^p$ is separable, by (Beck, 2017, Theorem 6.6) it is sufficient to deduce the characterization of $\text{prox}_{\lambda \| \cdot \|_p^p}$ in Eq. (3) when $MNC = 1$. Each component is thus given by

$$\left[ \text{prox}_{\lambda \| \cdot \|_p^p}(\boldsymbol{w}) \right]_i = \frac{2}{3} w_i \left( 1 + \cos \left( \frac{2\pi}{3} - \frac{2\phi_{2\lambda}(w_i)}{3} \right) \right) \mathbb{1}_S(i), \tag{4}$$

where

$$\phi_{2\lambda}(w_i) = \arccos \left( \frac{\lambda}{4} \left( \frac{|w_i|}{3} \right)^{-\frac{3}{2}} \right), \quad g(2\lambda) = \frac{\sqrt[3]{54}}{4} (2\lambda)^{\frac{2}{3}}, \quad S = \{ i \ : \ |w_i| > g(2\lambda) \}. \tag{5}$$

While the present solution to Eq. (2) for $p = \frac{1}{2}$ can generate highly sparse adversarial perturbations, these perturbations tend to be of large magnitude, making them easily perceptible (Fan et al., 2020). Moreover, this approach does not guarantee that the perturbations will affect the most critical pixels in the image. Our primary objective is to generate group-wise sparse adversarial attacks that are of low magnitude and targeted at the most important regions of the image.

## 2.2    GROUP-WISE SPARSE ADVERSARIAL ATTACKS OF LOW MAGNITUDE

We propose a two-phase method to generate meaningful group-wise sparse adversarial examples with minimal $\ell_2-$norm perturbation, and enhanced explainability as a natural byproduct (see Sec. 3.4.2).

**Step 1: Determine which pixel coordinates to perturb by tuning coefficients $\lambda$.** This step finds the most relevant group-wise sparse pixels to perturb by combining the $1/2-$quasinorm regularization

of Sec. 2.1 with a novel heuristic approach on $\lambda$. Instead of a single $\lambda$, we consider a vector of tradeoff parameters $\lambda \in \mathbb{R}_{\geq 0}^{M \times N \times C}$ for the $1/2-$quasinorm regularization term, allowing us to adjust each entry individually. Due to Eq. (4), we can formally define the proximal operator to a vector $\lambda \in \mathbb{R}_{\geq 0}^{M \times N \times C}$ of tradeoff parameters

$$\left[\text{prox}_{\lambda \|\cdot\|_p^p}(\boldsymbol{w})\right]_{i,j,c} := \left[\text{prox}_{\lambda_{i,j,c}\|\cdot\|_p^p}(\boldsymbol{w})\right]_{i,j,c}. \tag{6}$$

In the first $\hat{k}$ iterations, constituting Step 1, we reduce $\lambda_{i,j,:}$ for pixels located in proximity to already perturbed pixels (Eq. (10)), thereby making them more susceptible to perturbation.

**AdjustLambda.** After computing iterate $\boldsymbol{w}^{(k)}$ by forward-backward splitting with Nesterov momentum (line 4 in Algorithm 2), we update $\lambda^{(k)}$ by first building a mask

$$\boldsymbol{m} = \text{sign}\left(\sum_{c=1}^{C}|\boldsymbol{w}^{(k)}|_{:,:,c}\right) \in \{0,1\}^{M \times N}, \tag{7}$$

to identify the perturbed pixels. Next, we perform a 2D convolution on $\boldsymbol{m}$ using a square Gaussian blur kernel $\boldsymbol{K} \in \mathbb{R}^{n \times n}$ with appropriate padding, yielding a matrix

$$\boldsymbol{M} = \boldsymbol{m} * * \boldsymbol{K} \in [0,1]^{M \times N}, \tag{8}$$

where entries with indices close to non-zero entries of $\boldsymbol{m}$ are non-zero. Appendix B properly defines this operation. From the blurred perturbation mask $\boldsymbol{M}$, we compute a matrix $\overline{\boldsymbol{M}} \in \mathbb{R}^{M \times N}$ using

$$\overline{\boldsymbol{M}}_{ij} = \begin{cases} \boldsymbol{M}_{ij} + 1, & \text{if } \boldsymbol{M}_{ij} \neq 0, \\ q, & \text{else,} \end{cases} \tag{9}$$

for $0 < q \leq 1$. Finally, we increase (decrease) the tradeoff parameters for pixels situated far from (close to) those previously perturbed by computing new tradeoff parameters

$$\lambda_{i,j,:}^{(k+1)} = \frac{1}{\overline{\boldsymbol{M}}_{i,j}}\lambda_{i,j,:}^{(k)}, \quad \text{where } i \in \{1,...,M\} \text{ and } j \in \{1,...,N\}. \tag{10}$$

We thus expand regions with reduced tradeoff parameters over a specified $\hat{k}$ iterations, identifying the most relevant group-wise sparse pixels at coordinates $(i,j)$, where $\lambda_{i,j,c}^{(\hat{k})}$ is lower than the initial value $\lambda_{i,j,c}^{(0)}$. Unlike existing methods, ours operates independently of predefined structures like pixel partitions. This first step acts only as a heuristic to select the key group-wise sparse pixels, with the proper optimization problem formulated and solved over these selected coordinates in the next step.

**Step 2: Solve a low magnitude adversarial attack problem only over the selected coordinates.** In the remaining iterations, our algorithm formulates a simplified optimization problem over the pixel coordinates $V$ selected in Step 1 given by

$$\min_{\boldsymbol{w} \in V} \mathcal{L}(\boldsymbol{x} + \boldsymbol{w}, t) + \mu\|\boldsymbol{w}\|_2, \tag{11}$$

where $\mu > 0$ (same for all pixels) is a tradeoff parameter controlling $\ell_2-$norm perturbation magnitude. The subspace $V$ is defined as

$$V := \text{span}(\{e_{i,j,c} \mid \lambda_{i,j,c}^{(\hat{k})} < \lambda_{i,j,c}^{(0)}\}) \subseteq \mathbb{R}^{M \times N \times C}, \tag{12}$$

spanned by standard unit vectors $e_{i,j,c}$. The projection onto $V$ operates as

$$[P_V(\boldsymbol{w})]_{i,j,c} = \begin{cases} \boldsymbol{w}_{i,j,c}, & \text{if } e_{i,j,c} \in V, \\ 0, & \text{otherwise.} \end{cases} \tag{13}$$

Thus, the projection $P_V$ zeros out entries not in $V$ while leaving the others unchanged. We apply projected Nesterov's accelerated gradient descent (NAG) to solve Eq. (11), enabling generation of meaningful group-wise sparse perturbations of low magnitude. In the next iterations, perturbations only affect pixels at coordinates $(i,j)$ where $\lambda_{i,j,:}^{(\hat{k})}$ is less than the initial $\lambda$. For most perturbed pixels,

---

**Algorithm 2** GSE: Group-Wise Sparse and Explainable Attack

---
**Require:** Image $\boldsymbol{x} \in \mathcal{X}$, target label $t$, loss function $\mathcal{L}$, regularization parameters $\lambda, \mu > 0$, step size $\sigma > 0$, numbers of iterations $\hat{k}, K$, sequence $\alpha_k$.
1: Initialize $\boldsymbol{w}^{(0)} \leftarrow \boldsymbol{0}$, $\lambda^{(0)} \leftarrow \lambda\boldsymbol{1}$, define $f(\boldsymbol{w}) := \mathcal{L}(\boldsymbol{x} + \boldsymbol{w}, t) + \mu\|\boldsymbol{w}\|_2$
2: **for** $k \leftarrow 0, ..., \hat{k} - 1$ **do**
3:     $\tilde{\mathbf{w}}^{(k+1)} \leftarrow \mathrm{prox}_{\sigma\lambda^{(k)}\|\cdot\|_p^p} \left( \boldsymbol{w}^{(k)} - \sigma\nabla_{\boldsymbol{w}^{(k)}} \left( f(\boldsymbol{w}^{(k)}) \right) \right)$
4:     $\boldsymbol{w}^{(k+1)} \leftarrow (1 - \alpha_k)\tilde{\boldsymbol{w}}^{(k+1)} + \alpha_k\tilde{\boldsymbol{w}}^{(k)}$
5:     $\lambda^{(k+1)} \leftarrow \mathtt{AdjustLambda}(\lambda^{(k)}, \boldsymbol{w}^{(k+1)})$
6: **end for**
7: **for** $k = \hat{k}, ..., K - 1$ **do**
8:     $\tilde{\boldsymbol{w}}^{(k+1)} \leftarrow \boldsymbol{w}^{(k)} - \sigma\nabla_{\boldsymbol{w}^{(k)}} \left( f(\boldsymbol{w}^{(k)}) \right)$
9:     $\boldsymbol{w}^{(k+1)} \leftarrow P_V \left( (1 - \alpha_k)\tilde{\boldsymbol{w}}^{(k+1)} + \alpha_k\tilde{\boldsymbol{w}}^{(k)} \right)$          {Definition in Eq. (13).}
10: **end for**
11: **return** $\hat{\boldsymbol{w}} = \boldsymbol{w}^{(K)}$

---

all channels will be affected, as $e_{i,j,c} \in V$ across all channels $c$. This process is succinctly outlined in Algorithm 2, where we use a sequence $\alpha_k$ as in (Nesterov, 1983)

$$\beta_0 = 0, \ \ \beta_k = \frac{1}{2}\left(1 + \sqrt{1 + 4\beta_{k-1}^2}\right), \ \ \alpha_k = \frac{1}{\beta_{k+1}}\left(1 - \beta_k\right). \tag{14}$$

In our tests, we initially perform a section search to find the maximum $\lambda$ where $\tilde{\boldsymbol{w}}^{(1)} \neq \boldsymbol{0}$. Another section search then determines the appropriate $\lambda$ for a successful attack. The projected NAG in Algorithm 2 (solving Eq. (11)) converges as NAG solving an unconstrained problem, since the projection $P_V$ acts only within subspace $V$ (see Appendix C for proof).

## 3  EXPERIMENTS

We demonstrate the effectiveness of our proposed GSE attack for crafting group-wise sparse and explainable adversarial perturbations. After outlining experimental setup and defining the relevant comparing metrics, Sec. 3.4 compares our approach with multiple previous SOTA group-wise sparse adversarial attacks. Additionally, Sec. 3.4 covers ablation studies regarding the explainability of such attacks, visualizations, empirical time costs, and success against adversarially trained networks.

### 3.1  DATASETS

We experiment on CIFAR-10 (Krizhevsky et al., 2009) and ImageNet (Deng et al., 2009) datasets, analyzing DNNs on 10k randomly selected images from both validation sets. For the classifier $\mathcal{C}$ on CIFAR-10, we train a ResNet20 model (He et al., 2016) for 600 epochs using stochastic gradient descent, with an initial learning rate of 0.01, reduced by a factor of 10 after 100, 250, and 500 epochs. We set the weight decay to $10^{-4}$, momentum to 0.9, and batch size to 512. For ImageNet, we employ a ResNet50 (He et al., 2016) and a more robust transformer model, ViT_B_16 (Dosovitskiy et al., 2020), both with default weights from the torchvision library. All tests are conducted on a machine with an NVIDIA A40 GPU, and our codes, 10k image indices from the ImageNet validation dataset, and target labels for targeted ImageNet tests are available at https://github.com/wagnermoritz/GSE.

### 3.2  EVALUATION METRICS

Consider $n$ images $\boldsymbol{x}^{(1)}, ..., \boldsymbol{x}^{(n)}$ and the corresponding perturbations $\boldsymbol{w}^{(1)}, ..., \boldsymbol{w}^{(n)}$, where $\boldsymbol{x}^{(i)}, \boldsymbol{x}^{(i)} + \boldsymbol{w}^{(i)} \in \mathcal{X}$. Among these, let $m_s \leq n$ denote the number of successfully generated adversarial examples $\boldsymbol{x}^{(i)} + \boldsymbol{w}^{(i)}$. Then, the *attack success rate (ASR)* is simply defined as $\mathrm{ASR} = m_s/n$. Assume that the first $m_s$ adversarial examples are the successful ones. We compute the *average number of changed pixels (ACP)* via

$$\mathrm{ACP} = \frac{1}{m_s M N}\sum_{i=1}^{m_s}\|\boldsymbol{m}^{(i)}\|_0, \tag{15}$$

Table 1: Our attack vs. Homotopy-Attack and SAPF on a ResNet20 classifier for CIFAR-10. Perturbations for every attack were computed for all images sequentially. Due to the extensive computation time of Homotopy-Attack, we tested on a limited sample size of 100.

| Attack | Untargeted | | | | | | Targeted | | | | | |
|---|---|---|---|---|---|---|---|---|---|---|---|---|
| | ASR | ACP | ANC | $\ell_2$ | $d_{2,0}$ | Time | ASR | ACP | ANC | $\ell_2$ | $d_{2,0}$ | Time |
| GSE (Ours) | **100%** | **42.5** | **1.88** | 0.76 | **180** | **2.93s** | **100%** | **107** | **2.56** | 1.13 | **301** | **4.08s** |
| Homotopy | **100%** | 55.3 | 2.78 | 0.64 | 256 | 499s | **100%** | 113 | 4.27 | 1.05 | 394 | 754s |
| SAPF | **100%** | 87.3 | 4.59 | **0.41** | 324 | 250s | **100%** | 109 | 4.87 | **0.78** | 346 | 277s |

Table 2: Untargeted attacks on ResNet20 classifier for CIFAR-10, and ResNet50 and ViT_B_16 classifiers for ImageNet. Tested on 10k images of each dataset.

| | Attack | ASR | ACP | ANC | $\ell_2$ | $d_{2,0}$ |
|---|---|---|---|---|---|---|
| CIFAR-10 ResNet20 | GSE (Ours) | **100%** | **41.7** | **1.66** | **0.80** | **177** |
| | StrAttack | **100%** | 118 | 7.50 | 1.02 | 428 |
| | FWnucl | 94.6% | 460 | 1.99 | 2.01 | 594 |
| ImageNet ResNet50 | GSE (Ours) | **100%** | **1629** | 8.42 | **1.50** | **3428** |
| | StrAttack | **100%** | 7265 | 15.3 | 2.31 | 11693 |
| | FWnucl | 47.4% | 13760 | **3.79** | 1.81 | 16345 |
| ImageNet ViT_B_16 | GSE (Ours) | **100%** | **941** | **5.11** | **1.95** | **1964** |
| | StrAttack | **100%** | 3589 | 10.8 | 2.03 | 8152 |
| | FWnucl | 57.9% | 7515 | 5.67 | 3.04 | 9152 |

where the perturbation masks $m^{(i)}$ are defined as in Eq. (7). We utilize the ACP instead of the 0-norm since all tested group-wise sparse attacks minimize the ACP. To determine the number of perturbed pixel clusters in $x^{(i)} + w^{(i)}$, we create a mask $m^{(i)}$ as in Eq. (7) and treat it as a graph where adjacent 1-entries are connected nodes. The number of clusters in the perturbation is equal to the number of disjoint connected subgraphs, which we determine by depth-first search. Averaging this number over all successful examples gives us the *average number of clusters (ANC)*.

The value $\|w\|_{2,0}$ for a perturbation $w$, as suggested in (Zhu et al., 2021), heavily relies on the pixel partitioning method. Instead, we assess the group-wise sparsity of a perturbation by considering all $n_p$ by $n_p$ pixel patches of an image, rather than only a subset resulting from partitioning. Let $w \in \mathbb{R}^{M \times N \times C}$, $n_p < M, N$, and let $\mathcal{G} = \{G_1, ..., G_k\}$ be a set containing the index sets of all overlapping $n_p$ by $n_p$ patches in $w$. Then we define

$$d_{2,0}(w) := |\{i \ : \ \|w_{G_i}\|_2 \neq 0, \ i = 1, ..., k\}| . \tag{16}$$

Let $x$ and $w$ represent a vectorized image and its corresponding perturbation in $\mathbb{R}^d$, respectively. We measure the explainability of perturbations using the *interpretability score* (IS) (Xu et al., 2019a), derived from the *adversarial saliency map* (ASM) (Papernot et al., 2016b). For an image $x$ of true class $l$ and a target class $t$, $\text{ASM}(x, l, t) \in \mathbb{R}^d$ indicates the importance of each feature for classification. The IS is then defined as

$$\text{IS}(w, x, l, t) = \frac{\|\mathbf{B}(x, l, t) \odot w\|_2}{\|w\|_2}, \quad [\mathbf{B}(x, l, t)]_i = \begin{cases} 1, & \text{if } [\text{ASM}(x, l, t)]_i > \nu, \\ 0, & \text{otherwise,} \end{cases} \tag{17}$$

where $\nu$ is some percentile of the entries of $\text{ASM}(x, l, t)$. Note that, when $\text{IS}(w, x, l, t)$ approaches 1, the perturbation primarily targets pixels crucial for the class prediction of the model. For a visual representation of salient image regions, we use the *class activation map* (CAM) (Zhou et al., 2016). Recall that CAM identifies class-specific discriminative image regions, aiding in visually explaining adversarial perturbations. See Appendix D for more details on ASM, IS, and CAM.

We evaluate all attacks on these metrics in the untargeted and in the targeted setting. For CIFAR-10, the targeted attacks are performed with respect to each wrong label. For the evaluation of the attacks on ImageNet, we choose ten target labels for each image by randomly choosing ten distinct numbers $a_1, ..., a_{10} \in \{1, ..., 999\}$ and defining the target labels $t_i$ for an image with the true label $l$ by $t_i = l + a_i \mod 1000$. In the targeted setting, as suggested in (Fan et al., 2020), we give three versions of each metric: best, average, and worst case.

Table 3: Targeted attacks performed on ResNet20 classifier for CIFAR-10, and ResNet50 and ViT_B_16 classifiers for ImageNet. Tested on 1k images from each dataset, 9 target labels for CIFAR-10 and 10 target labels for ImageNet.

| | Attack | Best case | | | | | Average case | | | | | Worst case | | | | |
|---|---|---|---|---|---|---|---|---|---|---|---|---|---|---|---|---|
| | | ASR | ACP | ANC | $\ell_2$ | $d_{2,0}$ | ASR | ACP | ANC | $\ell_2$ | $d_{2,0}$ | ASR | ACP | ANC | $\ell_2$ | $d_{2,0}$ |
| CIFAR-10 ResNet20 | GSE (Ours) | **100%** | **29.6** | **1.06** | **0.68** | **137** | **100%** | **86.3** | **1.76** | **1.13** | **262** | **100%** | **162** | **3.31** | 1.57 | **399** |
| | StrAttack | **100%** | 78.4 | 4.56 | 0.79 | 352 | **100%** | 231 | 10.1 | 1.86 | 534 | **100%** | 406 | 15.9 | 4.72 | 619 |
| | FWnucl | **100%** | 283 | 1.18 | 1.48 | 515 | 85.8% | 373 | 2.52 | 2.54 | 564 | 40.5% | 495 | 4.27 | 3.36 | 609 |
| ImageNet ResNet50 | GSE (Ours) | **100%** | **3516** | 5.89 | 2.16 | **5967** | **100%** | **12014** | 14.6 | **2.93** | **16724** | **100%** | **21675** | **22.8** | **3.51** | **29538** |
| | StrAttack | **100%** | 6579 | 7.18 | 2.45 | 9620 | **100%** | 15071 | 18.0 | 3.97 | 20921 | **100%** | 26908 | 32.1 | 6.13 | 34768 |
| | FWnucl | 31.1% | 9897 | **3.81** | **2.02** | 11295 | 7.34% | 19356 | **7.58** | 3.17 | 26591 | 0.0% | N/A | N/A | N/A | N/A |
| ImageNet ViT_B_16 | GSE (Ours) | **100%** | **916** | **3.35** | 2.20 | **1782** | **100%** | **2667** | **7.72** | **2.87** | **4571** | **100%** | **5920** | **14.3** | **3.60** | **9228** |
| | StrAttack | **100%** | 3550 | 7.85 | **2.14** | 5964 | **100%** | 8729 | 17.2 | 3.50 | 13349 | **100%** | 16047 | 27.4 | 5.68 | 22447 |
| | FWnucl | 53.2% | 5483 | 4.13 | 2.77 | 6718 | 11.2% | 6002 | 9.73 | 3.51 | 7427 | 0.0% | N/A | N/A | N/A | N/A |

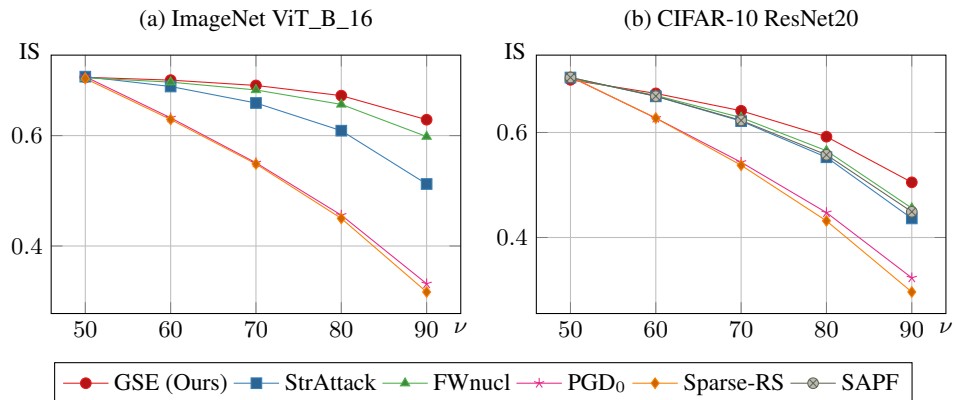

Figure 2: IS vs. percentile $\nu$ for targeted versions of GSE vs. five other attacks. Evaluated on an ImageNet ViT_B_16 classifier (a), and CIFAR-10 ResNet20 classifier (b). Tested on 1k images from each dataset, 9 target labels for CIFAR-10 and 10 target labels for ImageNet.

### 3.3 ATTACK CONFIGURATIONS

To find hyperparameters $q, \sigma, \mu$, and $\hat{k}$ of our algorithm, we run a grid search where ACP + ANC is the objective and ASR = 1.0 the constraint. The exact hyperparameters can be found in Appendix D.1. Regarding StrAttack, we modify the authors' implementation to be compatible with PyTorch, employing the parameters recommended in (Xu et al., 2019a), specifically those from Appendix F. For FWnucl, we implement the nuclear group norm attack in PyTorch, setting $\varepsilon = 5$. We adjust the implementation of Homotopy-Attack to support group-wise sparsity following (Zhu et al., 2021) and use their recommended parameter settings. For SAPF, we use hyperparameters from (Fan et al., 2020) and choose the sparsity parameter to minimize perturbed pixels while retaining a success rate of 100%. We run all the attacks for a total of 200 iterations.

### 3.4 RESULTS

As shown in Tab. 1, our method significantly outperforms the Homotopy-Attack and SAPF in both targeted and untargeted attacks on CIFAR-10. While Homotopy-Attack and SAPF surpass our method only slightly in the 2−norm perturbation magnitude, this metric is of secondary importance in the group-wise sparse attack setting. In contrast, our method excels in increasing the overall sparsity and group-wise sparsity, all while substantially reducing the cluster count and the computation time. Due to Homotopy-Attack's much slower speed and inability to process batches of images in parallel, we exclude it from further experiments. Additionally, as we couldn't replicate SAPF's ImageNet results due to floating-point overflow during the ADMM update, we also exclude SAPF. We proceed with the more recent FWnucl and StrAttack in further tests.

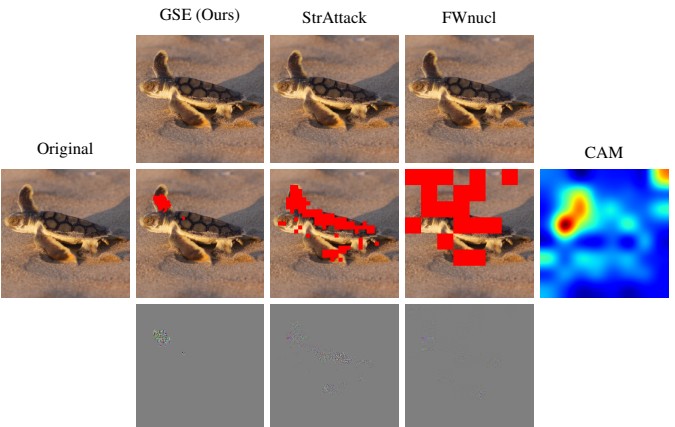

Figure 3: Visual comparison of successful untargeted adversarial instances generated by our attack, StrAttack, and FWnucl. Adversarial examples are shown in the top row, perturbed pixels highlighted in red in the middle row, and the perturbations in the bottom row. The target model is a ResNet50. Perturbations are enhanced for visibility.

### 3.4.1 EMPIRICAL PERFORMANCE

In Tab. 2 we present untargeted attack results on CIFAR-10 and ImageNet datasets. Notably, our method and StrAttack attain a 100% ASR in both cases. Furthermore, our algorithm significantly outperforms other attacks in terms of ACP. Specifically, our algorithm achieves an average sparsity of $4.1\%$ on CIFAR-10 and $0.5\%$ on ImageNet when attacking a ResNet50 classifier. Additionally, our method demonstrates exceptionally high group-wise sparsity, with an increase of $70.2\%$ on CIFAR-10 and $70.7\%$ on ImageNet when attacking a ResNet50 classifier, outperforming the current SOTA.

The results for targeted attacks are presented in Tab. 3, where our method and StrAttack achieve a perfect ASR in both cases. However, FWnucl results in a $0\%$ worst-case ASR on the ImageNet dataset indicating no worst-case results for FWnucl on ImageNet. Our algorithm yields the sparsest perturbations, changing on average only $8.4\%$ of the pixels in CIFAR-10 images and $13.4\%$ of the pixels in ImageNet images when attacking a ResNet50 classifier. Moreover, our method achieves remarkable group-wise sparsity, with a significant decrease of $d_{2,0}$ by $50.9\%$ on CIFAR-10 and $20\%$ on ImageNet when attacking a ResNet50 classifier, while maintaining the lowest magnitude of attacks measured by the $2-$norm. While GSE is outperformed by a SOTA attack (FWnucl) only in the number of clusters for ImageNet, this remains insignificant considering FWnucl's high ACP and low ASR compared to our method. Furthermore, replicating experiments with a ViT_B_16 classifier in Tab. 2 and Tab. 3 demonstrates our algorithm's substantial margin over SOTA, achieving a $73.8\%$ ($55.6\%$) increase in sparsity and $76\%$ ($38.4\%$) in group-wise sparsity in the untargeted (targeted) setting, while also attaining the lowest perturbation norm. Additional results can be found in Appendix E.1.

### 3.4.2 EXPLAINABILITY

Figure 2 illustrates the IS metric for the targeted attacks across various percentiles of ASM scores for samples from the CIFAR-10 and ImageNet datasets. Notably, the figure highlights that our method consistently achieves higher IS scores compared to other group-wise sparse attacks, especially for higher percentiles $\nu$. This shows that

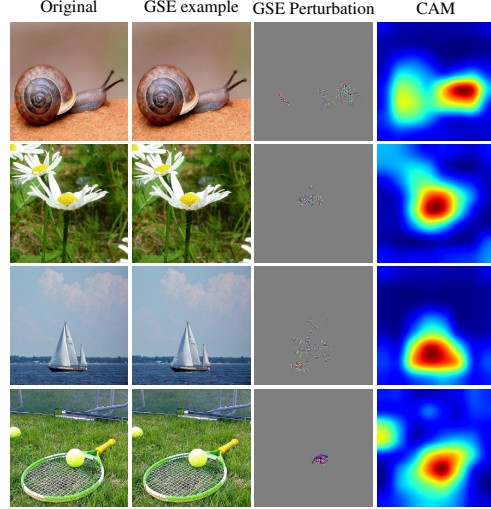

Figure 4: Targeted adversarial examples generated by GSE. The target is "airship" for the first two rows, and "golf cart" for the last two rows. The attacked model is a VGG19. Perturbations are enhanced for visibility.

Table 4: Comparison of empirical attack computation time per image. In every experiment, all attacks utilize a batch size of 100 images.

| | Untargeted | | | Targeted | | |
|---|---|---|---|---|---|---|
| | CIFAR-10 | ImageNet | | CIFAR-10 | ImageNet | |
| **Attack** | **ResNet20** | **ResNet50** | **ViT_B_16** | **ResNet20** | **ResNet50** | **ViT_B_16** |
| GSE (Ours) | **0.13s** | **3.18s** | **7.05s** | **0.16s** | **4.42s** | **10.3s** |
| StrAttack | 0.97s | 15.2s | 33.6s | 1.28s | 18.5s | 34.2s |
| FWnucl | 0.25s | 9.93s | 26.0s | 0.32s | 11.6s | 26.2s |

Table 5: Comparison of group-wise sparse adversarial attacks for an adversarially trained ResNet50 ImageNet classifier. Tested on 10k ImageNet instances.

| | **Attack** | **ASR** | **ACP** | **ANC** | $\ell_2$ | $d_{2,0}$ |
|---|---|---|---|---|---|---|
| ImageNet ResNet50 | GSE (Ours) | 98.5% | **14246** | 9.17 | 13.9 | **20990** |
| | StrAttack | **100%** | 16163 | 13.5 | **13.2** | 22097 |
| | FWnucl | 35.1% | 26542 | **8.83** | 15.6 | 25956 |

the perturbations generated by GSE are more focused on the most salient regions of the image. To emphasize group-wise sparsity's pivotal role in enhancing explainability, we include SOTA sparse adversarial attacks like $PGD_0$ (Croce & Hein, 2019) and Sparse-RS (Croce et al., 2022), all of which are outperformed by the group-wise sparse attack techniques considered. See Appendix E.2 for additional results.

We display the visualizations for untargeted group-wise sparse adversarial examples in Fig. 3, while Fig. 4 presents those for targeted group-wise sparse adversarial examples. Using the CAM technique (Zhou et al., 2016), we demonstrate the alignment between our algorithm's perturbations and localized, class-specific discriminative regions within the images. The generated perturbations effectively encompass the most discriminative areas of the objects, a testament to our algorithm's impressive achievement of explainability. In the perturbation depictions, unperturbed pixels are shown in grey since the perturbations are not required to be non-negative.

### 3.4.3 SPEED COMPARISON

Tab. 4 summarizes the runtime performance of our algorithm vs. SOTA methods. Notably, GSE exhibits significantly faster performance compared to both FWnucl and StrAttack. This speed advantage stems partially from the method used to enforce group-wise sparsity. While StrAttack calculates the Euclidean norm of each pixel group in every iteration and integrates it as regularization using ADMM, FWnucl computes a solution for a nuclear group-norm LMO in each iteration. In contrast, our attack initially computes a solution for the $1/2-$quasinorm proximal operator in the first $\hat{k}$ iterations. Subsequently, the attack transitions to projected NAG with $\ell_2-$norm regularization, which is less computationally costly than the methods employed by both StrAttack and FWnucl.

### 3.4.4 GSE AGAINST ADVERSARIALLY TRAINED NETWORKS

In this section, we present further results on the performance of GSE against an adversarially trained DNN, following the method outlined in (Madry et al., 2018), a standard approach for improving DNN robustness to adversarial attacks. Tab. 5 shows the results of group-wise sparse attacks on an adversarially trained ResNet50, using PGD projected onto a $2-$norm ball of radius 3, from MadryLab's Robustness package (Engstrom et al., 2019), tested on the ImageNet dataset. Overall, GSE shows superior tradeoffs, achieving competitive accuracy while altering the fewest pixels (ACP), maintaining a favorable ratio of perturbed pixels (ACP) to clusters (ANC), and recording the lowest $d_{2,0}$. Although FWnucl slightly outperforms GSE in ANC, this is due to its excessive number of perturbed pixels (ACP). StrAttack only marginally surpasses GSE in attack success rate (ASR) and $\ell_2-$norm. These results suggest that GSE generates perturbations that adversarially robust models struggle to defend against effectively.

## 4 CONCLUSION

We introduced "GSE", a novel algorithm for generating group-wise sparse and explainable adversarial attacks. Our approach is rooted in proximal gradient methods for non-convex programming, featuring additional control over changed pixels, and the use of projected NAG technique to solve optimization problems. Extensive experiments validate that GSE excels in producing group-wise sparse adversarial perturbations, all while simultaneously exhibiting the highest level of sparsity and the shortest computation time. Moreover, GSE excels over existing approaches in terms of quantitative metrics for explainability and offers transparency for visualizing the vulnerabilities inherent in DNNs. This endeavour not only establishes a new benchmark for the research community to evaluate the robustness of deep learning algorithms but also suggests a simple defense strategy: employing the adversarial examples generated by GSE in adversarial training (Madry et al., 2018). For more sophisticated solutions, we advocate further exploration. We anticipate no adverse ethical implications or future societal consequences from our research.

## ACKNOWLEDGEMENTS

This research was partially supported by the Deutsche Forschungsgemeinschaft (DFG, German Research Foundation) as part of Germany's Excellence Strategy through the Berlin Mathematics Research Center MATH+ (EXC-2046/1, Project ID: 390685689) and the Research Campus MODAL funded by the German Federal Ministry of Education and Research (BMBF grant number 05M14ZAM).

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

APPENDIX

## A  UNTARGETED ADVERSARIAL ATTACK FORMULATION

We can easily modify Eq. (1) to generate *untargeted adversarial attacks* by maximizing the loss $\mathcal{L}$ with respect to the true label $l$

$$\max_{\boldsymbol{w} \in \mathbb{R}^{M \times N \times C}} \mathcal{L}(\boldsymbol{x} + \boldsymbol{w}, l) - \lambda \mathcal{D}(\boldsymbol{w}), \tag{18}$$

where $\lambda > 0$ is a regularization parameter.

## B  GAUSSIAN BLUR KERNEL

Given $\boldsymbol{m} \in \{0,1\}^{M \times N}$ and the square Gaussian blur kernel $\boldsymbol{K} \in \mathbb{R}^{n \times n}$, the convolution matrix $\boldsymbol{m} * *\boldsymbol{K} \in [0,1]^{M \times N}$ is defined via

$$[\boldsymbol{m} * *\boldsymbol{K}]_{i,j} = \sum_{k=-\hat{n}}^{\hat{n}} \sum_{l=-\hat{n}}^{\hat{n}} \boldsymbol{K}_{k+\hat{n}+1, l+\hat{n}+1} \cdot \boldsymbol{m}_{i+k, j+l},$$

where $\hat{n} = \lfloor \frac{n}{2} \rfloor$, $\lfloor \cdot \rfloor$ is the floor function, and $n$ is an odd number.

In other words, Gaussian Blur kernel is a weighted mean of the surrounding pixels that gives more weight to the pixel near the current pixel.

Setting $\boldsymbol{M} = \boldsymbol{m} * *\boldsymbol{K}$ and computing the matrix $\overline{\boldsymbol{M}} \in \mathbb{R}^{M \times N}$ as in Eq. (9) via

$$\overline{\boldsymbol{M}}_{ij} = \begin{cases} \boldsymbol{M}_{ij} + 1, & \text{if } \boldsymbol{M}_{ij} \neq 0, \\ q, & \text{else,} \end{cases} \tag{19}$$

for $0 < q \leq 1$, means we increase the tradeoff parameters in Eq. (10) for pixels with a distance greater than $\hat{n} = \lfloor \frac{n}{2} \rfloor$ from any perturbed pixels.

## C  EQUIVALENCE OF PROJECTED NAG IN ALGORITHM 2 TO NAG OF UNCONSTRAINED PROBLEMS

Here we show that the projected NAG in Algorithm 2 has the same properties as NAG solving an unconstrained problem (Eq. (25)), and thus converges as such.

*Proof.* Consider $d = MNC$, where our image $\boldsymbol{x} \in [I_{\min}, I_{\max}]^d$ is vectorized. Let $\mathcal{I}$ denote the set that encompasses indices corresponding to entries with $\lambda_{i,j,c}^{(\hat{k})} \geq \lambda_{i,j,c}^{(0)}$ after $\hat{k}$ iterations. Also, let $m := |\mathcal{I}| < d$. With these considerations, we can formulate the optimization problem in Eq. (11), which arises following iteration $\hat{k}$, as

$$\begin{aligned} \min_{\boldsymbol{w} \in \mathbb{R}^d} \quad & \mathcal{L}(\boldsymbol{x} + \boldsymbol{w}, t) + \mu \|\boldsymbol{w}\|_2 \\ \text{s.t.} \quad & A\boldsymbol{w} = 0, \end{aligned} \tag{20}$$

where $A \in \{0,1\}^{m \times d}$ has rows

$$(\underbrace{0, ..., 0}_{i-1 \text{ times}}, 1, \underbrace{0, ..., 0}_{d-i \text{ times}}), \quad \forall i \in \mathcal{I}. \tag{21}$$

We can eliminate the equality constraints using the nullspace method. Let $H \in \mathbb{R}^{d \times d}$ be an orthogonal matrix and $R \in \mathbb{R}^{m \times m}$ an upper triangular matrix, obtained by QR-decomposition of $A^\top$, i.e.,

$$HA^\top = \begin{pmatrix} R \\ 0 \end{pmatrix}. \tag{22}$$

Further let $H = (Y, Z)^\top$, where $Y^\top \in \mathbb{R}^{m \times d}$ contains the first $m$ rows of $H$ and $Z \in \mathbb{R}^{d \times \{d-m\}}$. Because an orthogonal matrix $H$ possesses full rank, both $Y$ and $Z$ also exhibit full rank. Hence we can uniquely write any $w \in \mathbb{R}^d$ as

$$w = Y w_y + Z w_z = H^\top \begin{pmatrix} w_y \\ w_z \end{pmatrix}, \tag{23}$$

with $w_y \in \mathbb{R}^m, w_z \in \mathbb{R}^{d-m}$. In particular, for any $w \in \ker A$ we have

$$0 = Aw = AH^\top \begin{pmatrix} w_y \\ w_z \end{pmatrix} = (R^\top, 0) \begin{pmatrix} w_y \\ w_z \end{pmatrix} = R^\top w_y. \tag{24}$$

Given Eqs. (21) and (22), we can establish that both $A$ and $R$ possess rank $m$. Consequently, we can represent any $w \in \ker A$ as $Z w_z$, where $w_z \in \mathbb{R}^{d-m}$. Denoting the $w_z$ simply by $z$, we can formulate an unconstrained problem equivalent to Eq. (20)

$$\min_{z \in \mathbb{R}^{d-m}} \mathcal{L}(x + Zz, t) + \mu \|Zz\|_2. \tag{25}$$

Setting $f(w) = \mathcal{L}(x + w, t) + \mu \|w\|_2$ and $F(z) = f(Zz)$ we get from the update step of NAG

$$
\begin{aligned}
w_{k+1} &= Z z_{k+1} \\
&= Z \Big( (1 - \alpha_k) \big( z^{(k)} - \sigma \nabla F(z^{(k)}) \big) + \alpha_k \big( z^{(k-1)} - \sigma \nabla F(z^{(k-1)}) \big) \Big) \\
&= (1 - \alpha_k) \Big( w^{(k)} - \sigma Z Z^\top \nabla f(w^{(k)}) \Big) + \alpha_k \Big( w^{(k-1)} - \sigma Z Z^\top \nabla f(w^{(k-1)}) \Big) \\
&= (1 - \alpha_k) \Big( w^{(k)} - \sigma P_V(\nabla f(w^{(k)})) \Big) + \alpha_k \Big( w^{(k-1)} - \sigma P_V(\nabla f(w^{(k-1)})) \Big) \\
&= P_V \Big( (1 - \alpha_k) \big( w^{(k)} - \sigma \nabla f(w^{(k)}) \big) + \alpha_k \big( w^{(k-1)} - \sigma \nabla f(w^{(k-1)}) \big) \Big),
\end{aligned} \tag{26}
$$

where the last equality holds since $w^{(k)}, w^{(k-1)} \in V$. Thus, the projected NAG in Algorithm 2 shares the properties of NAG solving the unconstrained problem in Eq. (25), ensuring its convergence. $\quad\square$

## D  EXPLANATION OF ASM, IS, AND CAM EVALUATION METRICS

We provide further details on the ASM, IS, and CAM metrics as outlined in Sec. 3.2 of the paper.

Consider a vectorized image $x \in [I_{\min}, I_{\max}]^d$ with true label $l$ and target label $t$. Let $Z(x)$ represent the logits of a classifier. The *adversarial saliency map (ASM)* (Papernot et al., 2016b; Xu et al., 2019a) is defined as follows

$$
\begin{aligned}
[\text{ASM}(x, l, t)]_i &= \left( \frac{\partial Z(x)_t}{\partial x_i} \right) \left| \frac{\partial Z(x)_l}{\partial x_i} \right| \mathbb{1}_S(i), \\
S &= \left\{ i \in \{1, ..., d\} \ \middle| \ \frac{\partial Z(x)_t}{\partial x_i} \geq 0 \text{ or } \frac{\partial Z(x)_l}{\partial x_i} \leq 0 \right\}.
\end{aligned} \tag{27}
$$

The metric $\text{ASM}(x, l, t) \in \mathbb{R}_{\geq 0}^d$ provides a measure of importance for each pixel. It is worth noting that a higher ASM value signifies greater pixel significance. We compute a binary mask $\mathbf{B}(x, l, t) \in \{0, 1\}^d$ by

$$[\mathbf{B}(x, l, t)]_i = \begin{cases} 1, & \text{if } [\text{ASM}(x, l, t)]_i > \nu, \\ 0, & \text{otherwise}, \end{cases} \tag{28}$$

where $\nu$ is some percentile of the entries of $\text{ASM}(x, l, t)$. Given an adversarial perturbation $w \in \mathbb{R}^d$, we can now compute the *interpretability score (IS)* as

$$\text{IS}(w, x, l, t) = \frac{\|\mathbf{B}(x, l, t) \odot w\|_2}{\|w\|_2}. \tag{29}$$

Note that when $\text{IS}(\boldsymbol{w}, \boldsymbol{x}, l, t)$ approaches 1, the perturbation primarily targets pixels crucial for the class prediction of the model. Conversely, IS scores nearing zero do not lend themselves to meaningful interpretation based on ASM scores.

Let $\mathcal{C}$ be a convolutional neural network classifier and $f_k[i, j]$ be the activation of the unit $k$ at the coordinates $(i, j)$ in the last convolutional layer of $\mathcal{C}$ evaluated at $\boldsymbol{x}$. Then, using global average pooling after the last convolutional layer, the input to the softmax corresponding to label $l$ is

$$\sum_k w_k^l \sum_{i,j} f_k[i, j] = \sum_{i,j} \sum_k w_k^l f_k[i, j],$$

where $w_k^l$ are the weights corresponding to the label $l$ for unit $k$. Since $w_k^l$ indicate the importance of $\sum_{i,j} f_k[i, j]$ for class $l$, the *class activation map (CAM)* (Zhou et al., 2016) is defined by

$$[\text{CAM}_l]_{i,j} = \sum_k w_k^l f_k[i, j],$$

and directly indicates the importance of activation at $(i, j)$ for the classification of $\boldsymbol{x}$ as class $l$. To make the comparison of $\text{CAM}_l$ and $\boldsymbol{x}$ easier, the resulting class activation map is upscaled to the size of $\boldsymbol{x}$ using bicubic interpolation.

### D.1 HYPERPARAMETERS OF THE GSE ALGORITHM

To find $q, \sigma, \mu$, and $\hat{k}$, we run a grid search where ACP + ANC is the objective and ASR = 1.0 the constraint. Specifically, for CIFAR-10, we set $q = 0.25$, $\sigma = 0.005$, $\mu = 1$, and $\hat{k} = 30$, while for ImageNet, we use $q = 0.9$, $\sigma = 0.05$, $\mu = 0.1$, and $\hat{k} = 50$. Due to the significant efficiency of our attack compared to other group-wise sparse attacks, we can easily run a grid search to find appropriate hyperparameters when working with different datasets. The grid search code used to find the hyperparameters for CIFAR-10 and ImageNet is attached in https://github.com/wagnermoritz/GSE.

## E ADDITIONAL EXPERIMENTS

### E.1 EMPIRICAL PERFORMANCE

In this section, we present untargeted and targeted attack results on the CIFAR-10 dataset (Krizhevsky et al., 2009) using a WideResNet classifier (Zagoruyko & Komodakis, 2016) and ImageNet dataset (Deng et al., 2009) using a VGG19 classifier (Simonyan & Zisserman, 2015). Additionally, we include results on the NIPS2017 dataset (available at https://www.kaggle.com/competitions/nips-2017-defense-against-adversarial-attack/data), comprising 1k images of the same dimensionality ($299 \times 299 \times 3$) as the ImageNet dataset, on which we test the attacks against a VGG19 (Simonyan & Zisserman, 2015) and a ResNet50 (He et al., 2016) classifier. The results in Tab. 6 and Tab. 7 reinforce the findings of Sec. 3.4.1, highlighting GSE's ability to produce SOTA group-wise sparse adversarial attacks.

Table 6: Untargeted attacks on a WideResNet classifier for CIFAR-10, VGG19 classifier for ImageNet, and VGG19 and ResNet50 classifiers for NIPS2017. Tested on 10k images from the CIFAR-10 dataset, 10k images from the ImageNet dataset, and 1k images from the NIPS2017 dataset.

|  | Attack | ASR | ACP | ANC | $\ell_2$ | $d_{2,0}$ |
|---|---|---|---|---|---|---|
| CIFAR-10 WideResNet | GSE (Ours) | **100%** | **73.1** | **1.53** | 0.71 | **229** |
|  | StrAttack | **100%** | 78.9 | 4.76 | 1.01 | 289 |
|  | FWnucl | 97.6% | 303 | 1.87 | 2.28 | 438 |
|  | SAPF | **100%** | 95.1 | 3.96 | **0.51** | 292 |
| ImageNet VGG19 | GSE (Ours) | **100%** | **813.9** | 5.43 | **1.63** | **1855** |
|  | StrAttack | **100%** | 3035 | 8.57 | 1.78 | 6130 |
|  | FWnucl | 85.8% | 5964 | **2.46** | 2.73 | 7431 |
| NIPS2017 VGG19 | GSE (Ours) | **100%** | **973** | 6.65 | **1.49** | **2254** |
|  | StrAttack | **100%** | 3589 | 10.1 | 1.56 | 6487 |
|  | FWnucl | 90.1% | 6099 | **2.35** | 2.71 | 7701 |
| NIPS2017 ResNet50 | GSE (Ours) | **100%** | **1332** | 9.81 | **1.46** | **3036** |
|  | StrAttack | **100%** | 8211 | 17.1 | 2.79 | 13177 |
|  | FWnucl | 48.2% | 14571 | **5.01** | 1.77 | 17086 |

Table 7: Targeted attacks performed a WideResNet classifier for CIFAR-10, VGG19 classifier for ImageNet, and VGG19 and ResNet50 classifiers for NIPS2017. Tested on 1k images from each dataset, 9 target labels for CIFAR-10 and 10 target labels for ImageNet and NIPS2017.

| | Attack | Best case | | | | | Average case | | | | | Worst case | | | | |
|---|---|---|---|---|---|---|---|---|---|---|---|---|---|---|---|---|
| | | ASR | ACP | ANC | $\ell_2$ | $d_{2,0}$ | ASR | ACP | ANC | $\ell_2$ | $d_{2,0}$ | ASR | ACP | ANC | $\ell_2$ | $d_{2,0}$ |
| CIFAR-10 WideResNet | GSE (Ours) | **100%** | **49.8** | **1.04** | 0.50 | **141** | 94.8% | 112 | **1.45** | 0.81 | 238 | 85.2% | 234 | **2.71** | **1.25** | **388** |
| | StrAttack | **100%** | 61.7 | 2.32 | 0.82 | 244 | **98.8%** | 196 | 6.31 | 1.64 | 431 | **93.1%** | 398 | 11.3 | 3.01 | 579 |
| | FWnucl | **100%** | 210 | 1.12 | 1.56 | 324 | 82.1% | 269 | 2.11 | 2.06 | 390 | 44.2% | 365 | 3.56 | 3.58 | 472 |
| | SAPF | **100%** | 85.3 | 1.97 | **0.45** | 223 | 94.8% | **91.9** | 3.61 | 0.91 | 278 | 75.2% | **101** | 6.25 | 1.36 | 371 |
| ImageNet VGG19 | GSE (Ours) | **100%** | **1867** | 6.15 | **2.54** | **3818** | **100%** | **6580** | 14.5 | **3.28** | **10204** | **100%** | **15526** | 24.8 | **4.39** | **20846** |
| | StrAttack | **100%** | 4303 | 6.11 | 2.88 | 7013 | **100%** | 11568 | 16.7 | 3.59 | 17564 | **0.0%** | 21552 | 29.8 | 5.16 | 30451 |
| | FWnucl | 56.2% | 4612 | **1.51** | 2.90 | 6973 | 18.3% | 9134 | **2.53** | 3.65 | 12050 | 0.0% | N/A | N/A | N/A | N/A |
| NIPS2017 VGG19 | GSE (Ours) | **100%** | **2155** | 5.96 | 2.78 | **4617** | **100%** | **6329** | 15.4 | **3.44** | **11304** | **100%** | **15541** | 25.2 | **4.19** | **21581** |
| | StrAttack | **100%** | 4197 | 6.55 | **2.45** | 6847 | **100%** | 11326 | 18.5 | 3.66 | 17332 | **100%** | 21018 | 34.1 | 5.38 | 30103 |
| | FWnucl | 52.6% | 4543 | **1.82** | 2.63 | 5783 | 14.1% | 7517 | **2.92** | 3.81 | 11007 | 0.0% | N/A | N/A | N/A | N/A |
| NIPS2017 ResNet50 | GSE (Ours) | **100%** | **3505** | 6.31 | **2.51** | **5698** | **100%** | **9127** | 16.2 | **2.87** | **15704** | **100%** | **20247** | 27.5 | **3.12** | **26574** |
| | StrAttack | **100%** | 6344 | 6.92 | 2.54 | 9229 | **100%** | 15278 | 18.6 | 4.05 | 21090 | **100%** | 27922 | 33.7 | 6.51 | 35927 |
| | FWnucl | 30.2% | 9812 | **2.49** | 2.62 | 12845 | 11.8% | 11512 | **8.02** | 3.69 | 19493 | 0.0% | N/A | N/A | N/A | N/A |

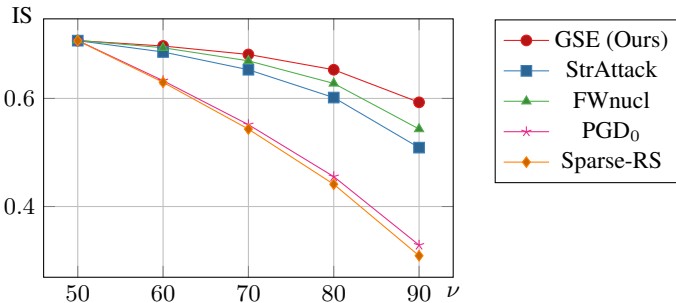

Figure 5: IS vs. percentile $\nu$ for targeted versions of GSE vs. four other attacks. Evaluated on an ImageNet VGG19 classifier. Tested on 1k images and 10 target labels for ImageNet.

### E.2 EXPLAINABILITY

Fig. 5 reinforces the findings of Sec. 3.4.2, demonstrating GSE's consistently higher IS scores compared to both SOTA group-wise sparse and SOTA sparse attacks, while using a VGG19 classifier (Simonyan & Zisserman, 2015) on ImageNet. This underscores that GSE-generated perturbations are more focused on the most salient regions of the image.

### E.3 TRANSFERABILITY

An intriguing application is evaluating the transferability performance (Papernot et al., 2016a) of various group-wise sparse attacks in the black-box setting, assessing whether an attack maintains a high ASR when targeting a different model. From Tab. 8, our GSE attack demonstrates transferability on par with SOTA group-wise sparse methods, outperforming StrAttack in the lower-triangular case and FWnucl in the diagonal case, while achieving a strong balance between in-model and out-model performance. The additional model, Swin-V2-B, is sourced from (Liu et al., 2022).

Table 8: Transferability of targeted attacks on ImageNet.

| Attack | Attacked Model | ResNet50 | Swin-V2-B | VGG19 |
|---|---|---|---|---|
| GSE | ResNet50 | 100% | 5.15% | 15.4% |
| | Swin-V2-B | 8.61% | 99.8% | 18.3% |
| | VGG19 | 5.01% | 4.26% | 100% |
| StrAttack | ResNet50 | 100% | 5.27% | 18.1% |
| | Swin-V2-B | 8.72% | 100% | 19.9% |
| | VGG19 | 4.95% | 4.12% | 100% |
| FWnucl | ResNet50 | 65.3% | 5.30% | 20.0% |
| | Swin-V2-B | 10.2% | 54.9% | 19.3% |
| | VGG19 | 9.62% | 6.05% | 97.5% |

Table 9: Ablation study on $\hat{k}$ for GSE. Tested on with 1,000 samples from the ImageNet dataset and a ResNet50 classifier. All values of $\hat{k}$ lead to an attack success rate of 100%.

| $\hat{k}$ | 10 | 20 | 30 | 40 | 50 | 60 | 70 | 80 | 90 | 100 |
|---|---|---|---|---|---|---|---|---|---|---|
| **ACP** | 1260 | 1342 | 1437 | 1399 | 1558 | 1519 | 1462 | 1450 | 1462 | 1433 |
| **ANC** | 16.3 | 13.4 | 10.5 | 9.69 | 8.11 | 6.73 | 5.90 | 5.14 | 4.92 | 4.36 |
| $d_{2,0}$ | 3529 | 3559 | 3442 | 3387 | 3356 | 3265 | 3048 | 2814 | 2773 | 2656 |
| $\ell_2$ | 1.44 | 1.42 | 1.41 | 1.39 | 1.43 | 1.41 | 1.42 | 1.40 | 1.43 | 1.44 |

### E.4 ABLATION STUDY ON $\hat{k}$

We conduct an ablation study on $\hat{k}$ to evaluate the impact of Step 1 on GSE attack performance, as shown in Tab. 9. We see that $\hat{k}$ is inversely correlated with the number of clusters (ANC) and group sparsity ($d_{2,0}$), as anticipated. No significant correlations with other metrics were observed.

