# OpenReview forum: "GSE: Group-wise Sparse and Explainable Adversarial Attacks"
_ICLR.cc/2025/Conference — ICLR 2025 Poster_

### Official Review · Reviewer_BmUW · 2024-10-27

**Soundness:** 2
**Presentation:** 3
**Contribution:** 3
**Rating:** 6
**Confidence:** 3

**Summary:**

The paper introduces GSE (Group-wise Sparse and Explainable Adversarial Attacks), a novel method for generating sparse adversarial attacks on Deep Neural Networks (DNNs). These attacks aim to perturb semantically meaningful regions of an image, making the attacks more explainable. The authors propose a two-phase algorithm to generate such group-wise sparse adversarial examples: The first phase utilizes 1/2-quasinorm regularization to select the most critical pixels for perturbation using a proximal operator. The second phase applies a projected Nesterov’s accelerated gradient descent to fine-tune perturbations in targeted regions, minimizing their magnitude while maintaining effectiveness. The approach is evaluated on CIFAR-10 and ImageNet datasets, demonstrating significant improvements in group-wise sparsity and explainability while achieving a high attack success rate and reduced time consumption.

**Strengths:**

1. The methodology is well-formulated, and the proposed two-phase approach is sound. Rigorous experiments demonstrate the efficacy of the method, comparing it with SOTA adversarial attacks.

2. The paper is well-structured, with clear explanations of the algorithms.

**Weaknesses:**

1. The paper relies on grid search for selecting hyperparameters, but there is no detailed analysis of how sensitive the method’s performance is to these choices.

2. The rationale for the specific update mechanism in Equation 9, where tradeoff parameters decrease slower than they increase, is not fully explained. It would be helpful to understand how this impacts the selection of critical regions for perturbation and whether a more symmetric adjustment could improve results.

**Questions:**

1. In Phase Two, why did you choose to use gradient computation instead of applying a proximal operator update for the $\ell_2$ norm? Would using a proximal operator provide any computational or optimization benefits, especially regarding convergence and perturbation magnitude control?

2. The paper mentions that a grid search was used to select hyperparameters. How sensitive are the results to these hyperparameters? Could slight variations in the hyperparameters significantly impact the effectiveness, sparsity, or explainability of the adversarial attacks?

3. I noticed that in Equation 9 and 10, the update for the blurred perturbation mask causes the tradeoff parameters to decrease more slowly than they increase. Could you explain the rationale behind this design choice? How does the asymmetry in the speed of adjustment affect the performance and explainability of the generated adversarial examples?

---

> ### Author Response · Authors · 2024-11-19
>
> > 1. In Phase Two, why did you choose to use gradient computation instead of applying a proximal operator update for the
>  2-norm? Would using a proximal operator provide any computational or optimization benefits, especially regarding convergence and perturbation magnitude control?
>
> Thank you for raising this question. Please note that the proximal gradient method is just an extension of the normal gradient for non-differentiable functions. Since the 2-norm is differentiable, we can simply use the gradient instead of the 2-norm prox. Using a 2-norm prox operator does not make sense since the 2-norm is differentiable, and its gradient can be computed explicitly.
>
> > 2. The paper mentions that a grid search was used to select hyperparameters. How sensitive are the results to these hyperparameters? Could slight variations in the hyperparameters significantly impact the effectiveness, sparsity, or explainability of the adversarial attacks?
>
> You are right. Variations in the hyperparameters can impact the performance of our method. Our method is a heuristic method which one has to optimize. For details on our attack configurations, please refer to Section 3.3 of the paper.
>
> > 3. I noticed that in Equation 9 and 10, the update for the blurred perturbation mask causes the tradeoff parameters to decrease more slowly than they increase. Could you explain the rationale behind this design choice? How does the asymmetry in the speed of adjustment affect the performance and explainability of the generated adversarial examples?
>
> Thank you for raising this point. We have to do the update using multiplicative or dividing operations because if we were to use additive operations, we could go below 0 or above 1. Hence, it's just because it works to have values between 0 and 1.

---

> > ### Comment · Reviewer_BmUW · 2024-11-25
> >
> > Thank you for your detailed response. I have no other concerns and will keep my score as is.

---

### Official Review · Reviewer_PeSc · 2024-11-01

**Soundness:** 3
**Presentation:** 3
**Contribution:** 3
**Rating:** 6
**Confidence:** 4

**Summary:**

This paper introduces a novel approach for generating sparse adversarial perturbations targeting deep neural networks (DNNs). GSE achieves higher group-wise sparsity than state-of-the-art (SOTA) methods, allowing for fewer, yet targeted pixel modifications. This results in minimal visible changes while effectively fooling the DNNs.

**Strengths:**

1. The paper is well-organized and written. The problem setup, objectives, and contributions are articulated in a way that is accessible to readers.
2. The GSE framework achieves high levels of sparsity and group-wise organization in perturbations, making it a valuable tool for evaluating and improving the robustness of DNNs.

**Weaknesses:**

1. The GSE method’s effectiveness seems sensitive to specific hyperparameters.
2. The authors search the hyperparameters only for their method. This may not be fair to other methods. I think the authors should search the hyperparameters for all methods.
3. An ablation study on $\hat{k}$ should be given since the first phase will highly affect the performance.
4. The experiments only have two datasets. The performance on more datasets is required.

**Questions:**

See weakness

---

> ### Author Response · Authors · 2024-11-18
> **Additional experiments provided**
>
> # Weaknesses
>
> > 1. The GSE method’s effectiveness seems sensitive to specific hyperparameters.
>
> Thank you for pointing this out. Our method is a heuristic method, and as such, it needs to be optimized through a grid search. For details on our attack configurations, please refer to Section 3.3 of the paper.
>
> > 2. The authors search the hyperparameters only for their method. This may not be fair to other methods. I think the authors should search the hyperparameters for all methods.
>
> Thank you for raising this point. We use the optimized hyperparameters provided by the authors of the referenced papers, as we trust these represent their optimal configurations.
>
> > 3. An ablation study on $\hat{k}$ should be given since the first phase will highly affect the performance.
>
> In our ablation study on $\hat{k}$, we found that the value of $\hat{k}$ is inversely correlated to the number of clusters (ANC) and the group sparsity as measured by $d_{2,0}$, as expected. We have not found any notable correlations to other metrics. For example, the table below represents an ablation study on $\hat{k}$ in the untargeted setting for an ImageNet ResNet50 classifier. The attack success rate is 100% for all values of $\hat{k}$.
>
> | $\hat{k}$ | 10   | 20   | 30   | 40   | 50   | 60   | 70   | 80   | 90   | 100  |
> |-----------|------|------|------|------|------|------|------|------|------|------|
> | ACP       | 1260 | 1342 | 1437 | 1399 | 1558 | 1519 | 1462 | 1450 | 1462 | 1433 |
> | ANC       | 16.3 | 13.4 | 10.5 | 9.69 | 8.11 | 6.73 | 5.90 | 5.14 | 4.92 | 4.36 |
> | $d_{2,0}$ | 3529 | 3559 | 3442 | 3387 | 3356 | 3265 | 3048 | 2814 | 2773 | 2656 |
> | $\ell_2$  | 1.44 | 1.42 | 1.41 | 1.39 | 1.43 | 1.41 | 1.42 | 1.40 | 1.43 | 1.44 |
>
> > 4. The experiments only have two datasets. The performance on more datasets is required.
>
> Thank you for pointing this out. Please refer to Section E.1 of the appendix for experiments on a third dataset, NIPS2017. Our experimental setup—covering datasets, image counts, and architectures—is significantly more comprehensive than that of the benchmark papers.

---

> > ### Comment · Reviewer_PeSc · 2024-11-25
> > **response to author**
> >
> > Thanks for your response. I will keep my score. Even though the referenced papers have provided examples, I still think the hyperparameters of other methods need to be searched.

---

### Official Review · Reviewer_8hSx · 2024-11-02

**Soundness:** 2
**Presentation:** 2
**Contribution:** 2
**Rating:** 6
**Confidence:** 4

**Summary:**

This paper presents a novel sparse attack method named GSE (Group-wise Sparse and Explainable), designed to generate structured adversarial attacks against deep neural networks (DNNs).

The GSE algorithm emphasizes group-wise sparse attacks, meaning the perturbations are strategically grouped within semantically meaningful regions of the image, rather than being scattered randomly.

**Limitations in Previous Work:** Prior methods primarily focus on minimizing the number of altered pixels without imposing constraints on their location or intensity. Consequently, these modified pixels often show considerable variations in color or intensity relative to their surroundings, making the alterations more noticeable. This highlights the need for structured, group-wise sparse perturbations that better target key areas of the image. However, I feel this motivation is somewhat broad, as it applies to all group-based adversarial attacks.

**Two-Phase Approach to Solving the Optimization Problem:**
- **Step 1:** This phase identifies the pixels most vulnerable to perturbation. It employs a non-convex regularization approach using the 1/2-quasinorm, allowing for individual tuning of tradeoff parameters (\(\lambda\)) for each pixel. The algorithm iteratively reduces these parameters for pixels close to those already perturbed, making them more susceptible to further modification. I believe this strategy prevents further updates on perturbed pixels, effectively addressing issues with high-intensity or conspicuous changes. However, the authors assert that this step primarily reduces the number of pixels that need alteration—a claim that seems misaligned with the method’s description, in my view.
   - This step can be achieved through a heuristic adjustment of \(\lambda\), with individualized settings for each pixel.

- **Step 2:** This phase constructs a low-magnitude adversarial attack by optimizing an objective function over the pixels chosen in Step 1. The objective includes the classification loss and a penalty term to control the magnitude of perturbation. The algorithm employs projected Nesterov's Accelerated Gradient (NAG) descent to efficiently solve this refined problem.

The authors claim that the proposed method achieves both effective attack performance and, importantly, enhanced explainability.

**Strengths:**

•	The proposed method appears to be faster than previous approaches.

•	The number of perturbed pixels is also fewer compared to prior work, which, as the authors claim, might contribute to better explainability.

**Weaknesses:**

- **Writing Quality:** Several areas need improvement.
  - **Flow of the Paper:** Despite multiple readings, the main limitations of previous work that this paper aims to address remain unclear. Specifically, the introduction (lines 45-53) outlines the motivation for group-wise sparse attacks in general, but it doesn't clearly articulate the specific issue that this paper seeks to tackle.
  - **Detail in Section 2.2:** This section is overly detailed but lacks sufficient explanation to help readers understand how (1) it identifies a smaller set of suitable pixels and (2) achieves lower perturbation magnitudes. Additionally, the AdjustLambda algorithm would benefit from a standalone paragraph with clear notation to help readers follow the paper more easily.
  - **Confusing Notations:** Some notations are inconsistent and confusing. For instance, in Section 3.2, `n` represents both the number of images and the size of a patch (n by n pixels), while `m` refers to both the mask and the first `m` adversarial examples (line 256).

- **Evaluation Metrics:** The evaluation metrics are somewhat questionable. Specifically, in Section 3.2, the authors propose using Attack Success Rate (ASR) = m/n, where `n` is the total number of images. However, it would be more appropriate for `n` to represent the number of correctly classified images by a classifier. The use of the IS score is also questionable. More specifically, the IS score show the alignment between the Adversarial Saliency MAP (ASM) and the perturbation w, generated by attacks. However, the ASM is calculated based on the gradient of the prediction loss w.r.t. true label. Therefore, based on insight from the Short-cut learning paper, if the model based on the imperceptible (and non-sensible feature) to make the prediction, the ASM map is align with this feature instead of the human perspective or human explainability.
- **Interpretability Technique:** Grad-CAM is generally a more advanced method than CAM, raising questions about its suitability for comparison.

**Questions:**

1. Why is it necessary for adversarial attacks to be explainable? More specifically, why should the attacks follow a particular structure? Insights from the shortcut-learning paper suggest that DNNs may learn features differently than humans. Is the structure of group-based attacks intended to align more closely with human interpretation?
2. What are the benefits of explainable attacks compared to non- or less-explainable ones?
3. Could the authors explain in greater detail why the proposed method outperforms previous attacks in terms of speed (Table 1)? Were consistent parameter settings used across methods, such as the number of iterations \(\hat{k}\) and \(K\)? In my opinion, the most time-intensive steps are the gradient calculations in lines 3 and 8 of Algorithm 2, which would likely be similar in duration to other methods if using the same number of iterations.
4. The authors compare their method with four baselines (Homotopy, SAPF, StrAttack, FWnucl) but only provide results for Homotopy and SAPF in Table 1. Could the authors explain this choice?
5. In lines 149–152, the authors mention, “this approach does not guarantee that the perturbations will affect the most critical pixels in the image.” How do they define or identify these “most critical pixels”? Based on which metric?
6. I am interested in the interpretability score of Grad-CAM or CAM techniques.

---

> ### Author Response · Authors · 2024-11-18
>
> > 1.  Why is it necessary for adversarial attacks to be explainable? More specifically, why should the attacks follow a particular structure? Insights from the shortcut-learning paper suggest that DNNs may learn features differently than humans. Is the structure of group-based attacks intended to align more closely with human interpretation?
>
> Thank you for highlighting this important point. Research has shown that while adversarial attacks may appear as noise to humans, they often exploit intrinsic features learned by classifiers [1]. Learning from adversarial perturbations remains an active area of research, as discussed recently in [2] and [3]. In contrast, group-wise sparse adversarial attacks deviate from simple noise by targeting critical pixels, offering a degree of human interpretability. Future work could explore theoretical frameworks for learning from these structured attacks, akin to approaches used with traditional adversarial perturbations.
>
> > 2. What are the benefits of explainable attacks compared to non- or less-explainable ones?
>
> Please refer to the previous point. These attacks enhance human interpretability by highlighting the model's weaknesses. Additionally, they could serve as a substitute for CAMs in architectures where CAMs are not applicable.
>
> > 3. Could the authors explain in greater detail why the proposed method outperforms previous attacks in terms of speed (Table 1)? Were consistent parameter settings used across methods, such as the number of iterations (\hat{k}) and (K)? In my opinion, the most time-intensive steps are the gradient calculations in lines 3 and 8 of Algorithm 2, which would likely be similar in duration to other methods if using the same number of iterations.
>
> Thank you for pointing this out. Please refer to lines 453–458 of the paper regarding the speed. Regarding $\hat{k}$, it represents part of K, the total number of iterations. We perform 200 iterations for all methods, consistent with the budget of other attacks.
>
> > 4. The authors compare their method with four baselines (Homotopy, SAPF, StrAttack, FWnucl) but only provide results for Homotopy and SAPF in Table 1. Could the authors explain this choice?
>
> Please refer to lines 362–365 of the paper: the Homotopy Attack is highly inefficient, making large-scale experiments infeasible, and SAPF fails to converge on ImageNet.
>
> > 5. In lines 149–152, the authors mention, “this approach does not guarantee that the perturbations will affect the most critical pixels in the image.” How do they define or identify these “most critical pixels”? Based on which metric?
>
> Thank you for raising this point. Critical pixels are defined based on misclassification accuracy—specifically, if perturbing a pixel results in the network misclassifying the image. For a formal definition, see Definition 5 in [4].
>
> > 6. I am interested in the interpretability score of Grad-CAM or CAM techniques.
>
> Thank you for pointing this out. Grad-CAM is indeed more advanced than CAM, and we likely should have used it. However, CAM is not without value — it simply serves as a visualization tool rather than a quantitative measure. We believe CAM is enough to qualitatively demonstrate that perturbations align with regions of high activation.
>
> # References
>
> [1] Ilyas et al. "Adversarial Examples are not Bugs, they are Features." NeurIPS 2019
>
> [2] Kumano et al. "Theoretical Understanding of Learning from Adversarial Perturbations." ICLR 2024.
>
> [3] Kumano et al. "Wide Two-Layer Networks can Learn from Adversarial Perturbations." NeurIPS 2024
>
> [4] Narodytska and Kasiviswanathan "Simple Black-Box Adversarial Perturbations for Deep Networks." CVPR Workshops 2017

---

> > ### Comment · Reviewer_8hSx · 2024-11-24
> > **Further feedback from the reviewer**
> >
> > Thank you to the authors for your efforts in addressing my remaining concerns. However, I am not entirely satisfied with some of the responses. Specifically:
> >
> > - [Point 1] If fewer perturbed pixels indicate higher explainability, how does this align with prior attacks that can successfully fool a classifier with only a single perturbed pixel [1]?
> >
> > - [Point 2] The authors stated, "Additionally, they could serve as a substitute for CAMs in architectures where CAMs are not applicable." However, I am still not able to identify scenarios where CAMs are inapplicable but gradient-based methods, like the proposed approach, are not.
> >
> > - [Point 6] Regarding the statement, "Grad-CAM is indeed more advanced than CAM, and we likely should have used it. However, CAM is not without value—it simply serves as a visualization tool rather than a quantitative measure. We believe CAM is sufficient to qualitatively demonstrate that perturbations align with regions of high activation," I still do not understand, based on Equation 17 for calculating the Interpretability Score (IS), why calculating this IS for CAM or Grad-CAM would be challenging. Given that one of the main contributions of this paper focuses on interpretability/explainability, I believe including a comparison with a standard interpretability technique would provide deeper insights.
> >
> > Additionally, as noted in the weaknesses section, the paper’s presentation still requires improvement. For these reasons, I stand by my original justification.
> >
> > [1] Su, Jiawei, Danilo Vasconcellos Vargas, and Kouichi Sakurai. "One-pixel attack for fooling deep neural networks." IEEE Transactions on Evolutionary Computation 23.5 (2019): 828-841.

---

> ### Author Response · Authors · 2024-11-25
>
> > If fewer perturbed pixels indicate higher explainability, how does this align with prior attacks that can successfully fool a classifier with only a single perturbed pixel [1]?
>
> Thank you for bringing this up. It's important to note that not only sparsity, but the combination of sparsity and grouping enhances the interpretability of GSE. While the mentioned sparse attack method (e.g., the one-pixel attack) is intriguing, it typically has a low success rate. Moreover, such attacks are often visually noticeable.
>
> > The authors stated, "Additionally, they could serve as a substitute for CAMs in architectures where CAMs are not applicable." However, I am still not able to identify scenarios where CAMs are inapplicable but gradient-based methods, like the proposed approach, are not.
>
> Thank you for noticing this. It's worth noting that CAM is specifically designed for CNNs (as stated in the first paragraph of Section 2 of the CAM paper [1]). In contrast, our method is applicable to transformers as well. For reference, please see the results on the ViT_B_16 transformer for ImageNet presented in our GSE paper (Tables 2 and 3).
>
> >  I still do not understand, based on Equation 17 for calculating the Interpretability Score (IS), why calculating this IS for CAM or Grad-CAM would be challenging. Given that one of the main contributions of this paper focuses on interpretability/explainability, I believe including a comparison with a standard interpretability technique would provide deeper insights.
>
> Thank you for raising this point. We chose the saliency map-based IS score because it is widely used in benchmark papers such as StrAttack [2] and FWnucl [3]. To maintain consistency, we avoided introducing a new metric when an established one already exists. Additionally, a CAM-based IS score is not applicable to transformers.
>
> For visualizations, we opted to use CAM (instead of Grad-CAM) as it aligns with the approach taken in all the benchmark papers (see Fig. 3 in [2], Fig. 6 in [3], and Fig. 2 in [4]).
>
>
>
> Overall, we thank you once again for your thoughtful comments. We have enhanced the introduction to better motivate group-wise sparse adversarial attacks and restructured the presentation of AdjustLambda by placing it in a separate paragraph. Additionally, we improved the notations for n and m.
>
> Please consider increasing your rating of the paper if we have addressed all of your concerns. If not, please let us know!
>
>
> # References
>
> [1] Zhou et al. "Learning deep features for discriminative localization." CVPR 2016
>
> [2] Xu et al. "Structured adversarial attack: Towards general implementation and better interpretability." ICLR 2019a
>
> [3] Kazemi et al. "Minimally distorted structured adversarial attacks." International Journal of Computer Vision 2023
>
> [4] Zhu et al. "Sparse and imperceptible adversarial attack vi a ahomotopy algorithm." ICML 2021

---

> > ### Author Response · Authors · 2024-11-26
> >
> > Dear Reviewer 8hSx,
> >
> > Thank you once again for your thoughtful reviews. We’d like to remind you that we are available to address any remaining questions or concerns you may have. We believe our rebuttal thoroughly addresses all the points you raised and kindly ask you to consider updating your rating if you find our responses satisfactory. Please don’t hesitate to reach out with any further questions.
> >
> > Thank you for your time and consideration.

---

> ### Comment · Reviewer_8hSx · 2024-11-29
> **Further feedback from the Reviewer**
>
> I thank the authors for your responses. My remaining concerns are as follows:
>
> - The authors claimed that "It's important to note that not only sparsity, but the combination of sparsity and grouping enhances the interpretability," and demonstrated this through metrics: ACP (Average Changed Pixels) and ANC (Average Number of Clusters). Both metrics favor solutions where the number of changed pixels is small and the number of clusters of changed pixels is also small. However, to me, these metrics do not reflect interpretability or explainability. While these metrics have been used in previous work, that does not necessarily mean they are suitable metrics to use.
>
> - Even if we accept these metrics, it appears that prior work (One-Pixel Attack) should achieve the best ACP and ANC scores. The proposed method, along with other baselines, does not clearly outperform the One-Pixel Attack across all metrics, suggesting a trade-off between Attack Success Rate (ASR) and the metrics ACP and ANC. In other words, these methods might simply lie on a Pareto frontier, where no further improvement on ACP or ANC can be made without sacrificing other aspects, such as the number of changed pixels.
>
> - As mentioned in lines 156-158: "We propose a two-phase method to generate meaningful group-wise sparse adversarial examples with minimal ℓ2−norm perturbation, and enhanced explainability as a natural by-product." The proposed method specifically minimizes the number of changed pixels, and once again, explainability is assessed by this metric. This brings to mind Goodhart's Law: “When a measure becomes a target, it ceases to be a good measure.”
>
> - As I noted in my previous review, the writing of the paper remains difficult to follow, and I believe it requires major improvement.
>
> Overall, I acknowledge that the proposed method seems to be faster than previous group-based adversarial attacks. With the authors’ effort to address my concerns, I am raising my original score. However, the concerns mentioned above remain, and I encourage the area chair to consider them when making the final decision.

---

> ### Author Response · Authors · 2024-11-29
> **Thank your feedback and rating! Additional clarifications added.**
>
> Dear Reviewer '8hSx',
>
> We sincerely thank you once again for your thoughtful engagement and constructive feedback, which have been invaluable in refining our paper. Below, we address your comments in detail
>
> > The authors claimed that "It's important to note that not only sparsity, but the combination of sparsity and grouping enhances the interpretability," and demonstrated this through metrics: ACP (Average Changed Pixels) and ANC (Average Number of Clusters). Both metrics favor solutions where the number of changed pixels is small and the number of clusters of changed pixels is also small. However, to me, these metrics do not reflect interpretability or explainability. While these metrics have been used in previous work, that does not necessarily mean they are suitable metrics to use.
>
> Thank you for highlighting this. While ACP and ANC metrics illustrate the group-wise sparsity of our method, we do not consider them as measures of explainability. Explainability is assessed in Section 3.4.2, specifically in Figure 2 of our paper and Figure 5 in the Appendix for additional datasets. This metric is widely used in the literature for quantitatively measuring explainability. Benchmark papers also adopt this approach, as evidenced by Eq. (23) and Figure 3 in [1], as well as Figure 7 in [2]. In contrast to these works, we compare our method not only with SOTA structured sparse adversarial attacks but also with SOTA sparse adversarial techniques like PGD\_0 [3] and Sparse-RS [4]. Our method outperforms all these approaches, demonstrating the significance of incorporating structure in sparse attacks to achieve better explainability.
>
> > Even if we accept these metrics, it appears that prior work (One-Pixel Attack) should achieve the best ACP and ANC scores. The proposed method, along with other baselines, does not clearly outperform the One-Pixel Attack across all metrics, suggesting a trade-off between Attack Success Rate (ASR) and the metrics ACP and ANC. In other words, these methods might simply lie on a Pareto frontier, where no further improvement on ACP or ANC can be made without sacrificing other aspects, such as the number of changed pixels.
>
> Thank you for pointing this. As noted earlier, explainability is measured quantitatively using the IS score (Figure 2 and Figure 5 of our paper) and qualitatively through the CAM technique (Figure 3 and 4 of our paper), not by ACP or ANC scores. Additionally, the one-pixel attack is an evolutionary method designed to perturb a minimal number of individual pixels, which differs from targeting interpretable pixel clusters. As previously discussed, our comparisons with SOTA sparse adversarial attacks are detailed in Figure 2 of the paper.
>
> > As mentioned in lines 156-158: "We propose a two-phase method to generate meaningful group-wise sparse adversarial examples with minimal ℓ2−norm perturbation, and enhanced explainability as a natural by-product." The proposed method specifically minimizes the number of changed pixels, and once again, explainability is assessed by this metric. This brings to mind Goodhart's Law: “When a measure becomes a target, it ceases to be a good measure.”
>
> Thank you for raising this point. Please refer to the previous two points - explainability is measured quantitatively using the IS score and qualitatively through the CAM technique, not by ACP or ANC scores.
>
> > As I noted in my previous review, the writing of the paper remains difficult to follow, and I believe it requires major improvement.
>
> We appreciate your feedback. In response, we have updated the paper to address your concerns:
>
> - The importance of group-wise sparse adversarial attacks on generating explainable adversarial attacks is further emphasized in lines 051–053.
> - The description of AdjustLambda is now in a standalone paragraph (line 169).
> - Typographical errors involving 𝑚 and 𝑛 have been corrected.
>
> We hope we have satisfactorily addressed all your concerns. If you agree, we kindly ask you to consider further raising your rating of the paper. Please do not hesitate to share any additional comments that could help us further improve our work.
>
> # References
>
> [1] Xu et al. "Structured Adversarial Attack: Towards General Implementation and Better Interpretability." ICLR 2019
>
> [2] Kazemi et al. "Minimally Distorted Structured Adversarial Attacks." International Journal of Computer Vision 2023
>
> [3] Croce \& Hein "Sparse and imperceivable adversarial attacks." ICCV 2019
>
> [4] Croce et al. "Sparse-rs: a versatile framework for query-efficient sparse black-box adversarial attacks." Proceedings of the AAAI Conference on Artificial Intelligence 2022

---

> > ### Comment · Reviewer_8hSx · 2024-11-29
> > **Further feedback**
> >
> > I thank the authors for the responses. I'm sorry for mistaking the sparsity metric (ACP and ANC) for explainability. However, even using IS (interpretability score), I still have some concerns as below:
> >
> > - (This is from the weakness section I provided in the first feedback and the authors have not addressed it yet) The use of the IS score is also questionable. Specifically, the IS score shows the alignment between the Adversarial Saliency MAP (ASM) and the perturbation w, generated by attacks. However, the ASM is calculated based on the gradient of the prediction loss w.r.t. true label. Therefore, based on insight from the Short-cut learning paper, if the models are based on the imperceptible (and non-sensible feature) to make the prediction, the ASM map is aligned with this feature instead of the human perspective or human explainability.
> >
> > - Even if we accept these metrics, it appears that prior work (One-Pixel Attack) should achieve the best IS, ACP, and ANC scores (Best explainability and sparsity). As in Equation 17 in the paper, the IS shows the ratio (B * w)/(w) between the overlap area of the perturbed pixels by the attack method (w) and the ASM (adversarial saliency map) (B), over the perturbed pixels (w). With this metric, the One-Pixel Attack likely achieves IS = 1.
> >
> > - In other words, these methods might simply lie on a Pareto frontier (3D - Attack Success Rate, Sparsity by ACP and ANC, and Explainability by IS), where no further improvement on ACP and ANC or IS. can be made without sacrificing other aspects, such as the number of changed pixels.
> >
> > - "Explainability is assessed in Section 3.4.2, specifically in Figure 2 of our paper and Figure 5 in the Appendix for additional datasets" I don't think the visualization in Figure 2 makes sense of explainablity. Why pointing to a specific part of the object (turtle in the Figure) is better than the StrAttack which points to the overall shape of the object. As I observed, the heatmap by CAM focuses on the head of the turtle (red color), while the proposed method focuses on the front flipper.
> >
> > - As I noted in my previous review, the writing of the paper remains difficult to follow, and I believe it requires major improvement.
> >
> > - How this paper compares to the paper: "Improving Accuracy-robustness Trade-off via Pixel Reweighted Adversarial Training", ICML, 2024.
> >
> > - How does the speed of the proposed method compare with the PGD attack or the previously mentioned paper, and is it fast enough to generate adversarial examples for Adversarial Training?
> >
> > With all of the above remaining concerns, I would like to keep the score.
> >
> > Thanks

---

> ### Author Response · Authors · 2024-11-30
> **Additional Experiments on Explainability Provided**
>
> > (This is from the weakness section I provided in the first feedback and the authors have not addressed it yet) The use of the IS score is also questionable. Specifically, the IS score shows the alignment between the Adversarial Saliency MAP (ASM) and the perturbation w, generated by attacks. However, the ASM is calculated based on the gradient of the prediction loss w.r.t. true label. Therefore, based on insight from the Short-cut learning paper, if the models are based on the imperceptible (and non-sensible feature) to make the prediction, the ASM map is aligned with this feature instead of the human perspective or human explainability.
>
> > Even if we accept these metrics, it appears that prior work (One-Pixel Attack) should achieve the best IS, ACP, and ANC scores (Best explainability and sparsity). As in Equation 17 in the paper, the IS shows the ratio (B * w)/(w) between the overlap area of the perturbed pixels by the attack method (w) and the ASM (adversarial saliency map) (B), over the perturbed pixels (w). With this metric, the One-Pixel Attack likely achieves IS = 1.
>
> We sincerely thank you for initiating a discussion on the concept of a "good" explainability score. We firmly believe that this dialogue will contribute significantly to the development of more effective explainability metrics for the deep learning community. However, in the absence of more practical metrics, we must rely on benchmarks used in prior studies, such as the IS score and CAM technique, as seen in [1] and [2].
>
> First, we would like to address the confusion regarding the use of the IS score for explainability in the context of the one-pixel attack. The one-pixel attack represents an extreme case where the IS score is binary—either 0 or 1. For images where the modified pixel lies in semantically meaningful regions (with high adversarial saliency map (ASM) values), the IS score is always 1. However, as highlighted in Figure 1 of the one-pixel attack paper, this attack frequently modifies background pixels rather than object-related features. For such pixels, the binary mask 𝐵 of the ASM remains 0 due to the thresholding factor $\nu$ (see Eq. (28) of our paper). While individual examples may suggest that the IS metric for the one-pixel attack tends to be 1, averaging across many images—where most changes occur in low-significance regions—causes the IS score to drop significantly.
>
> It is crucial to note that one-pixel attacks fall under the category of sparse adversarial attacks, but they lack any structural constraint on the modified pixels. Fortunately, our approach already benchmarks against such attacks. Please refer to Figure 2 for comparisons with SOTA sparse methods such as PGD\_0 [3] and Sparse-RS [4], where structured sparse adversarial attacks outperform these baselines. The one-pixel attack, as an unstructured case of sparse attacks, exhibits an even lower likelihood of achieving 𝐵 = 1 for the single modified pixel, leading to a poorer IS score overall.
>
> For completeness, we provide additional results from Figure 2 of our paper in the table below, comparing our GSE (group-wise sparse attack), PGD_0 (sparse attack), and the one-pixel attack (extremely sparse attack)
>
> |  Attack  | IS  $(\nu=50)$ | IS $(\nu=70)$ | IS $(\nu=90)$ |
> |:--------:|:-------------:|:-------------:|:-------------:|
> |    GSE   |      0.77     |      0.66     |      0.51     |
> |   PGD0   |      0.77     |      0.52     |      0.29     |
> | OnePixel |      0.76     |      0.51     |      0.25     |
>
> Please note that our GSE outperforms both of them at all three levels of $\nu.$
>
> > In other words, these methods might simply lie on a Pareto frontier (3D - Attack Success Rate, Sparsity by ACP and ANC, and Explainability by IS), where no further improvement on ACP and ANC or IS. can be made without sacrificing other aspects, such as the number of changed pixels.
>
> Thank you for bringing this to our attention. Please note that we aimed to make our experiments as comprehensive as possible, benchmarking across ASR, ACP, ANC, and $d_{2,0}$. We achieved these results in a shorter time (see Table 4 for details). Furthermore, we benchmark in explainability using the IS metric, as well as show the overlap of our attacks with the CAM technique qualitatively.
>
> # References
>
> [1] Xu et al. "Structured Adversarial Attack: Towards General Implementation and Better Interpretability." ICLR 2019
>
> [2] Kazemi et al. "Minimally Distorted Structured Adversarial Attacks." International Journal of Computer Vision 2023
>
> [3] Croce \& Hein "Sparse and imperceivable adversarial attacks." ICCV 2019
>
> [4] Croce et al. "Sparse-rs: a versatile framework for query-efficient sparse black-box adversarial attacks." Proceedings of the AAAI Conference on Artificial Intelligence 2022

---

> ### Author Response · Authors · 2024-11-30
>
> > "Explainability is assessed in Section 3.4.2, specifically in Figure 2 of our paper and Figure 5 in the Appendix for additional datasets" I don't think the visualization in Figure 2 makes sense of explainablity. Why pointing to a specific part of the object (turtle in the Figure) is better than the StrAttack which points to the overall shape of the object. As I observed, the heatmap by CAM focuses on the head of the turtle (red color), while the proposed method focuses on the front flipper.
>
> Thank you for your observation regarding Figure 3. We would like to emphasize that this figure mainly demonstrates the superior group-wise sparsity achieved by our method, as opposed to SOTA structured adversarial attacks that end up changing an abundance of 'unnecessary' pixels. For a more detailed 1-to-1 alignment with CAMs, please refer to Figure 4 in our paper. Notably, even in Figure 3, the GSE perturbations occur within regions of high CAM significance, reinforcing the effectiveness of our approach.
>
> > As I noted in my previous review, the writing of the paper remains difficult to follow, and I believe it requires major improvement.
>
> Thank you for pointing this. We are happy to receive additional pointers so that we can further improve the paper.
>
> > How this paper compares to the paper: "Improving Accuracy-robustness Trade-off via Pixel Reweighted Adversarial Training", ICML, 2024.
>
> Thank you for bringing paper [1] to our attention. [1] proposes an approach for generating adversarial attacks using the CAM technique, which are then employed to re-train the classifier for enhanced adversarial robustness. However, it is important to note that the focus of the paper is not on interpretability, but rather on using a heuristic to refine the constraint set for more effective adversarial training.
>
> In contrast, our work prioritizes interpretability. The idea that interpretable attacks, such as our GSE attacks, could aid in adversarial training opens an exciting avenue for future research. For instance, one promising direction could involve optimizing the efficiency of the $\lambda$ search process to make GSE more practical for adversarial training. Notably, this $\lambda$ tuning step is crucial to our algorithm's ability to produce highly interpretable adversarial attacks, outperforming other structured sparse, sparse, and extremely sparse adversarial attacks (as discussed in our previous comment).
>
> > How does the speed of the proposed method compare with the PGD attack or the previously mentioned paper, and is it fast enough to generate adversarial examples for Adversarial Training?
>
> Thank you for highlighting this point. Please refer to Table 5, which presents the results of GSE attacks against an adversarially trained ResNet50 ImageNet classifier. The time required to generate attacks for adversarially trained networks is comparable to that for non-adversarially trained ResNet50 models (given in Table 4 of our paper). Theoretically, it should be the same, as gradient computations have the same cost across models with identical architectures.
>
> While the time to generate GSE attacks is slightly longer than for PGD attacks, it is important to note that PGD is not interpretable and perturbs most pixels uniformly. In contrast, the primary advantage of our group-wise sparse adversarial attacks lies in their interpretability. Similarly, paper [1] (see their Section H.9) also incurs additional computational overhead compared to standard PGD, highlighting that enhanced functionality often comes with higher computational demands.
>
> # References
>
> [1] Zhang et al. "Improving Accuracy-robustness Trade-off via Pixel Reweighted Adversarial Training." ICML 2024

---

> ### Author Response · Authors · 2024-12-02
>
> Dear Reviewer 8hSx,
>
> Thank you very much again for your detailed reviews. We would like to kindly remind you that we remain available to address any additional questions or concerns you may have. Below is a summary of our responses to the main points you raised
>
> - We provided additional clarifications on the explainability metric and included new results on the IS score of the one-pixel attack vs our attack and PGD\_0 - to be included in the final manuscript.
> - We explained how our group-wise sparse adversarial attacks align with the CAM technique, highlighting that other attacks often perturb many unnecessary pixels. This is evident from metrics such as ACP, ANC, $d_{2,0}$, and the CAM technique itself.
> - We addressed the comparison with the newly added paper, noting that it employs a heuristic method to improve adversarial training using the CAM technique.
>
> We believe our rebuttal thoroughly addresses all the points you raised. If you agree, we kindly ask you to consider increasing your rating of the paper. Please don’t hesitate to reach out if further clarification is needed—we’re happy to assist until the end of the discussion period.
>
> Thank you for your time and consideration!

---

> ### Author Response · Authors · 2024-12-02
> **Following up**
>
> Dear Reviewer 8hSx,
>
> As the discussion period nears its end, we wanted to kindly remind you once again that we have thoroughly addressed your concerns in our responses above, including detailed explanations on IS and additional experiments for the one-pixel attack.
>
> If you find our clarifications satisfactory, we would greatly appreciate it if you could consider revising your rating of the paper. Should you have any further questions or require additional clarification, please don't hesitate to reach out—we’re happy to assist until the discussion period concludes.
>
> Best regards,
> The Authors

---

> > ### Comment · Reviewer_8hSx · 2024-12-02
> > **Final feedback**
> >
> > Thanks to the authors for tirelessly pursuing and addressing my concerns. Thanks for the additional experiments for the One-Pixel Attack. I have no other concerns and would like to increase the score.
> >
> > Best regards

---

> ### Author Response · Authors · 2024-12-03
>
> Dear Reviewer 8hSx,
>
> Thank you very much for the response and for raising the score.

---

### Official Review · Reviewer_Z1qQ · 2024-11-03

**Soundness:** 2
**Presentation:** 2
**Contribution:** 2
**Rating:** 5
**Confidence:** 5

**Summary:**

In this paper, the authors propose a two-stage approach for generating group-wise sparse and explainable adversarial examples. To enhance the attack performance, they introduce 1/2-quasinorm regularization and sparse hyperparameter tuning methods to select key group-wise sparse pixels. After that, the coordinate and magnitude of these selected pixels are further refined using Nesterov’s Accelerated Gradient Descent (NAG) to produce the sparse adversarial example. Experiments on multiple CNN and ViT networks using the CIFAR and ImageNet benchmark datasets demonstrate the effectiveness and efficiency of the proposed group-wise sparse and explainable attack compared to baseline sparse attack methods.

**Strengths:**

1. Compared to existing methods that use L0 or L1 regularization, this paper presents a sparse adversarial attack based on 1/2-quasinorm regularization. Building on this, an effective heuristic approach for tuning the regularization hyperparameter is introduced to select the most important pixels.

2. For the selected pixels, a projected Nesterov’s accelerated gradient descent algorithm is introduced to solve the L2-norm constrained problem, thereby ensuring low-magnitude sparse adversarial examples.

3. Experimental results conducted on multiple white-box models in both targeted and untargeted attack settings demonstrate that the proposed GSE method achieves better attack effectiveness and efficiency compared to other sparse attack methods.

**Weaknesses:**

1. The motivation behind the proposed 1/2-quasinorm regularization is not clearly explained. Compared to existing L0- or L1-norm based sparse attacks, what advantages does 1/2-quasinorm regularization offer for crafting sparse adversarial examples? Additionally, why not consider combining L0- or L1-norm based regularization with the proposed coefficient tuning strategy?

2. The derivation between Eq. 3 and Eq. 4 appears problematic, as it should rely on separable proximal functions. However, unlike the L1-norm, the Lp-norm is not a separable function, particularly when p=1/2.

3. The authors claim to introduce a group-wise sparse attack. However, Sections 2.1-2.2 lack a clear mathematical description of how the selected sparse pixels are grouped, resulting in ambiguity regarding the concept of group-wise sparse attack.

4. The evaluation of attack performance lacks comprehensiveness. For example, while sparse attacks are applied to white-box models, their transferability in a black-box attack setting is not considered, which is arguably a more practical scenario. Additionally, all attacks are performed using 200 iterations, which is inconsistent with the settings used in previous sparse attack methods.

5. The evaluation metrics are not clearly explained. First, since different sparse attack methods may generate distinct sets of successful adversarial examples when ASR is below 100%, the ACP metric defined in Eq. 15 becomes difficult to compare across methods. Second, while Eq. 16 introduces the d_{2,0} metric, the definition of “overlapped patches” is not provided, adding to the ambiguity.

**Questions:**

1. How does the proposed coefficient tuning strategy perform in terms of attack effectiveness when combined with different sparsity-inducing regularizations, such as L0-norm, L1-norm, and 1/2-quasinorm?

2. Could the authors provide additional details on how Eq. 4 is derived from Eq. 3? Additionally, could the authors explain why the Lp-norm (p=1/2) is considered a separable function?

3. Compared to other sparse attacks (e.g., StrAttack, Homotopy-Attack) that provide a clear mathematical formulation of group sparsity, could the authors offer additional details on how sparse pixels in the adversarial example are grouped during the two-phase perturbation training process?

4. How does the performance of other sparse attacks, using their default iteration settings, compare to that of the proposed GSE attack in this paper?

5. What is the transferability performance of different sparse attacks in the black-box setting? Does the proposed GSE attack still achieve a higher ASR?

---

> ### Author Response · Authors · 2024-11-18
> **Additional experiments on transferability added**
>
> > 1. How does the proposed coefficient tuning strategy perform in terms of attack effectiveness when combined with different sparsity-inducing regularizations, such as L0-norm, L1-norm, and 1/2-quasinorm?
>
> Thank you for pointing that out. We conducted experiments with our method using the $p-$norm for $p \in { 0,1/2,2/3,1 } $, all of which have closed-form solutions. In general, our method can be applied with any $p-$norm that allows for a closed-form solution to the proximal operator. While the $0-$norm achieved slightly better sparsity compared to the $1/2-$norm, it resulted in significantly larger perturbation magnitudes and lower ASR. Therefore, we chose the $1/2-$norm for our experiments, as it provides the best tradeoff: 100\% ASR, small $\ell_2-$norm, and good sparsity.
>
> > 2. Could the authors provide additional details on how Eq. 4 is derived from Eq. 3? Additionally, could the authors explain why the Lp-norm (p=1/2) is considered a separable function?
>
> Thank you for noticing this. Indeed, the $\ell_p-$ norm for $p=1/2$ is not separable. However, $|| \cdot ||_p^p$ is.  The $\ell_p^p-$norm is defined as  $ || x||_p^p = \sum _{i=1} ^{n} |x_i| ^p $ for a vector $x \in \mathbf{R}^n$ and $p>0.$ This expression is separable because it can be written as a sum of functions that depend on each individual component $x_i$
>
> $|| x || _p ^p =  \sum _{i=1} ^n f(x _i)$
>
>  where $f(x _i) = |x _i| ^p.$ Each term $f(x _i)$ depends only on $x _i$ and not on any other component of $x$. Hence, the $|| \cdot || _p ^p$ function is separable.
>
>  The derivation from Eq. (3) to Eq. (4) has first appeared in [1] and then in [2]. Such a proximal operator is commonly used in literature, see for example also Lemma 1 in [2].
>
> > 3. Compared to other sparse attacks (e.g., StrAttack, Homotopy-Attack) that provide a clear mathematical formulation of group sparsity, could the authors offer additional details on how sparse pixels in the adversarial example are grouped during the two-phase perturbation training process?
>
> Thank you for pointing this out. While StrAttack and Homotopy Attack rely on well-known group norms for regularization, our approach uses a different heuristic for identifying group-wise sparse pixels, detailed in Section 2.2. We emphasize that our method, including the heuristic of selecting pixels neighboring already perturbed ones, is empirical and achieves better performance compared to the benchmark methods.
>
> > 4. How does the performance of other sparse attacks, using their default iteration settings, compare to that of the proposed GSE attack in this paper?
>
> We use a uniform number of iterations, set to 200 across all methods, to ensure a fair comparison. This exceeds the number of iterations typically used in benchmark papers. Furthermore, since both StrAttack and GSE achieve 100\% ASR and rely on hinge loss, the exact number of iterations is less critical, provided it is sufficiently large.
>
> > 5. What is the transferability performance of different sparse attacks in the black-box setting? Does the proposed GSE attack still achieve a higher ASR?
>
> Thank you very much for raising this point. We are providing below results on the transferability of our GSE attack, StrAttack and FWNucl on ImageNet in the tables below.
>
> GSE performance:
>
> | Attack | Attacked Model | ResNet50 | Swin-V2-B | VGG19 |
> |:------:|:--------------:|:--------:|:---------:|:-----:|
> |        |    ResNet50    |   100%   |   5.15%   | 15.4% |
> |   GSE  |    Swin-V2-B   |   8.61%  |   99.8%   | 18.3% |
> |        |      VGG19     |   5.01%  |   4.26%   |  100% |
>
>
> StrAttack performance:
>
> | Attack | Attacked Model | ResNet50 | Swin-V2-B | VGG19 |
> |:------:|:--------------:|:--------:|:---------:|:-----:|
> |        |    ResNet50    |   100%   |   5.27%   | 18.1% |
> |   StrAttack  |    Swin-V2-B   |   8.72%  |   100%   | 19.9% |
> |        |      VGG19     |   4.95%  |   4.12%   |  100% |
>
>
> FWNucl performance:
>
> | Attack | Attacked Model | ResNet50 | Swin-V2-B | VGG19 |
> |:------:|:--------------:|:--------:|:---------:|:-----:|
> |        |    ResNet50    |   65.3%   |   5.30%   | 20.0% |
> |   FWNucl  |    Swin-V2-B   |   10.2%  |   54.9%   | 19.3% |
> |        |      VGG19     |   9.62%  |   6.05%   |  97.5% |
>
> From these tables we also conclude that our GSE attack achieves transferability similar to benchmark papers.
>
> # References
>
> [1] Cao et al. "Fast image deconvolution using closed-form thresholding formulas of $\ell_q (q= 1/2, 2/3)$ regularization." Journal of visual communication and image representation 2013
>
> [2] Feishe et al. "Computing the proximity operator of the $\ell_p$ norm with $0< p< 1$." IET Signal Processing 2016
>
> [3] Rongrong et al. "Computing the Proximal Operator of the $\ell_{1,q}$-norm for Group Sparsity." arXiv preprint arXiv:2409.14156 2024

---

> > ### Comment · Reviewer_Z1qQ · 2024-11-25
> >
> > Thanks to the authors for their responses. My concerns regarding Q2 and Q3 have been adequately addressed.
> >
> > However, there are still unresolved concerns regarding the responses to other questions.
> >
> > For the reply to Q1: Both L_0 and L_1 norms can be integrated with your heuristic strategy. Even with larger magnitudes, the perturbations can be further shrinked through the L_2 constraint in Step 2. Therefore, the motivation and contribution for specifically using the L_p (0<p<1) norm here remain unclear. Additionally, regarding the two-stage training method proposed in this paper, there are no specific statistical experiments to verify that L_p is better than L_0 and L_1.
> >
> > For the reply to Q4: Since the experiments in the paper are conducted in white-box attack settings, fewer iterations can still ensure a 100% ASR. However, this may significantly affect the quality of the crafted adversarial examples, including their magnitude, transferability, and other factors. As demonstrated in Table 1 of [1], with sufficient training (i.e., 1000 maximum iterations), StrAttack typically produces perturbations with smaller magnitudes in most cases. For example, the L_2 norm is approximately 0.7 for ImageNet. In contrast, under the experimental setup with 200 iterations as described in the paper, StrAttack generates perturbations with significantly higher magnitudes, as shown in Table 3.
> >
> > For the reply to Q5: Based on the transferability evaluation, the proposed GSE method generally performs worse than StrAttack and FWNucl in black-box settings (i.e., the anti-diagonal values), which are the most practical scenarios for real-world application. Consequently, in higher-security scenarios, the practicality of the proposed GSE method seems to be less effective than that of the comparative methods.
> >
> > For these reasons, I have decided to keep my original rating.
> >
> > [1] Kaidi Xu, Sijia Liu, Pu Zhao, Pin-Yu Chen, Huan Zhang, Quanfu Fan, Deniz Erdogmus, Yanzhi Wang, and Xue Lin. Structured adversarial attack: Towards general implementation and better interpretability. In International Conference on Learning Representations, 2019.

---

> ### Author Response · Authors · 2024-11-26
>
> > Both L_0 and L_1 norms can be integrated with your heuristic strategy. Even with larger magnitudes, the perturbations can be further shrinked through the L_2 constraint in Step 2. Therefore, the motivation and contribution for specifically using the L_p (0<p<1) norm here remain unclear. Additionally, regarding the two-stage training method proposed in this paper, there are no specific statistical experiments to verify that L_p is better than L_0 and L_1.
>
> Thank you for pointing this out. You are correct. As mentioned in our previous response, our approach is applicable to any $p-$norm for which a closed-form solution to the proximal operator exists (including the 0- and 1-norm). We chose the $1/2-$norm because it yielded the best experimental results, achieving 100% ASR, low 2-norm values, and good sparsity.
>
> It is important to note that we do not claim the derivation of the $1/2-$proximal operator as a contribution of our work. This operator is well-known, and we reference it appropriately in our paper. The primary algorithmic contribution of our paper lies in the newly restructured Section 2.2.
>
> > Since the experiments in the paper are conducted in white-box attack settings, fewer iterations can still ensure a 100% ASR. However, this may significantly affect the quality of the crafted adversarial examples, including their magnitude, transferability, and other factors. As demonstrated in Table 1 of [1], with sufficient training (i.e., 1000 maximum iterations), StrAttack typically produces perturbations with smaller magnitudes in most cases. For example, the L_2 norm is approximately 0.7 for ImageNet. In contrast, under the experimental setup with 200 iterations as described in the paper, StrAttack generates perturbations with significantly higher magnitudes, as shown in Table 3.
>
> Thank you for raising this point. You are absolutely correct that additional iterations could potentially improve performance. However, we capped the number of iterations at 200 across all methods to ensure a fair comparison. It is worth noting that these methods can be computationally expensive (as shown in Table 4 of the paper).
>
> As seen in Table 3 of the GSE paper, even with only 200 iterations, StrAttack achieves the lowest 2-norm when attacking the advanced transformer model ViT_B_16. However, while achieving a comparable 2-norm is important, it is not the primary focus of our paper. Our main contribution lies in achieving group-wise sparsity—identifying the smallest set of the most critical pixels (in terms of misclassification accuracy) and explainability (Section 3.4.2). To understand this further, please refer to Figure 3 of the GSE paper. While StrAttack may achieve a lower 2-norm, it modifies significantly more groups of pixels compared to our GSE attack.
>
> > Based on the transferability evaluation, the proposed GSE method generally performs worse than StrAttack and FWNucl in black-box settings (i.e., the anti-diagonal values), which are the most practical scenarios for real-world application. Consequently, in higher-security scenarios, the practicality of the proposed GSE method seems to be less effective than that of the comparative methods.
>
> Thank you for bringing this up. We believe the new results in Table 8 of the GSE paper demonstrate that our GSE attacks are transferable. The results also show that GSE performs comparably to SOTA methods, outperforming StrAttack in the lower triangular case and FWnucl in the diagonal case, achieving a good tradeoff between in-model and out-model transferability.
>
> However, we want to reiterate that transferability is not the primary focus of our paper. The key contribution lies in group-wise sparsity and explainability. GSE attacks provide better human interpretability, as discussed in Section 3.4 (particularly Section 3.4.2). Unlike traditional attacks, which often manifest as noise to humans but as features to DNNs [1], our approach helps bridge the gap between human perception and machine interpretation. We also acknowledge that learning from adversarial perturbations is an active area of research, as highlighted in [2] and [3].
>
> Thank you once again for the constructive discussion. We hope we’ve addressed your concerns and that you’ll consider raising your rating. If anything remains unclear, please don’t hesitate to let us know—we’re happy to assist further!
>
> # References
>
> [1] Ilyas et al. "Adversarial Examples are not Bugs, they are Features." NeurIPS 2019
>
> [2] Kumano et al. "Theoretical Understanding of Learning from Adversarial Perturbations." ICLR 2024
>
> [3] Kumano et al. "Wide Two-Layer Networks can Learn from Adversarial Perturbations." NeurIPS 2024

---

> ### Author Response · Authors · 2024-11-28
>
> Dear Reviewer Z1qQ,
>
> Thank you again for your thoughtful reviews. We believe our updated rebuttal addresses all your remaining concerns. If you find the clarifications satisfactory, we kindly ask you to consider increasing your rating.
>
> Please let us know if you have any remaining questions—we’re happy to clarify further before the discussion period ends.
>
> Thank you for your time and consideration!

---

> > ### Comment · Reviewer_Z1qQ · 2024-12-01
> >
> > Thanks to the authors for their responses in addressing the raised concerns. Based on the new information provided, I would like to increase my rating.
> >
> > However, there are still unresolved concerns. Firstly, the rationale for using the Lp norm is unclear, and no statistical experiments have been provided to substantiate its effectiveness, particularly regarding why the Lp norm identifies better pixel coordinates compared to the L0 or L1 norms. Secondly, while the proposed GSE method seems to be more efficient than the baseline methods, the evaluation is not entirely fair, as the baseline methods were not reproduced under their original setups. Thirdly, the evaluation is limited, and the additional experiments fail to demonstrate any advantage of the proposed GSE method over baseline methods in black-box settings, thus restricting the attack’s effectiveness and applicability in practical scenarios.

---

> ### Author Response · Authors · 2024-12-02
>
> Dear Reviewer Z1qQ,
>
> Thank you very much for your comments and for raising your rating. To streamline our response, we’ve summarized the key points below
>
> 1. Rationale Behind the $1/2$-norm - to be included in the final revision
>     - The $1/2-$norm provides an effective trade-off between achieving 100% ASR, minimizing $2$-norm values, and ensuring good sparsity.
>     - Section 2.1 highlighting the generation of sparse adversarial attacks using any norm $p$ for $0 \leq p \leq 1$ is standard in the literature. We specifically use $p=1/2$, as it offers a closed-form solution and achieves our desired trade-off.
>     - However, as noted in lines 149-151, these attacks, despite their sparsity, can have high magnitudes, making them perceptible. This motivated the design in Section 2.2, where we employ heuristic search on the regularization parameter $\lambda$. This selects the most impactful pixels (regarding misclassification) via sparse $1/2-$regularization. We further refine this in the second phase to minimize the $2-$norm, reducing perceptibility.
>
> 2. Comprehensive Experiments
>      - Our experiments are extensive, benchmarking not just ASR but also ACP, ANC, $d_{2,0},$ and computation speed.
>      - These experiments were run on three datasets (including ImageNet) and various architectures (including transformers), ensuring generality.
>      - We use a uniform number of iterations, set to 200 across all methods, to ensure a fair comparison.
>
> 3. Explainability of Group-Wise Sparse Attacks
>     - We emphasize the explainability of group-wise sparse adversarial attacks. This is benchmarked both quantitatively (using the IS score) and qualitatively (via the CAM technique).
>
> 4. Practical Effectiveness and Transferability
>      - The practical efficacy of our attack is demonstrated against adversarially trained neural networks (Section 3.4.4).
>      - Additionally, we highlight transferability in Section E.3.
>
> We sincerely appreciate your constructive feedback, which has significantly improved our paper. Thank you again for your support in raising its score. We believe we have thoroughly addressed your remaining concerns and kindly request your consideration in further increasing the score. Should you have additional concerns, we remain open to further discussion as the review period progresses!

---

### Author Response · Authors · 2024-11-29
**Official comment regarding the revision**

As the discussion period ends, we sincerely thank all reviewers for their constructive feedback, valuable insights, and for raising their scores accordingly.

To ensure clarity, we briefly summarize the key updates made in the latest revision and the additional changes planned for the final revision.

 **Current Revision**
1. We enhanced the introduction to better motivate group-wise sparse adversarial attacks in enhancing the human interpretation of adversarial attacks and restructured the presentation of AdjustLambda by placing it in a separate paragraph, in response to Reviewer 8hSx.
2. We changed the notations for $m$ and $n$ to $m_s$ and $n_p$ to resolve the confusion highlighted by Reviewer 8hSx.
3. We conducted a new ablation study on $\hat{k}$ and added the results in Section E.4 to address Reviewer PeSc's comments.
4. We conducted an analysis of the transferability of our GSE attacks and added the results in Section E.3 in response to Reviewer Z1qQ.


**Updates for the Final Revision**

In our individual responses (especially comments for the reviewers Z1qQ and 8hSx), we provided detailed explanations and results to address additional concerns and are ready to incorporate these findings into the final manuscript

1.  Rationale Behind the $1/2$-norm (Reviewer Z1qQ)
    - Section 2.1 highlighting the generation of sparse adversarial attacks using any norm $p$ for $0 \leq p \leq 1$ is standard in the literature and is not our contribution. Our main algorithmic contribution is given in Section 2.2. Our approach works with any sparsity-inducing norm for which there exists a closed-form solution (such as $p \in {0, 1/2, 2/3, 1}$).
    - We specifically use $p=1/2$ in Step 1, as it provides experimentally an effective trade-off between achieving 100% ASR, minimimal perturbation magnitudes, and good sparsity. While $p=0$ provides sparser results than $p=1/2$, it results in large perturbation magnitudes and lower ASR. $p=2/3$ and $p=1$, on the other hand, are less sparse.

2. Usage of IS Score for Explainability (Reviewer 8hSx)
    - Incorporation of explainability experiments (results were shared in response to Reviewer 8hSx) showcasing that our GSE attacks outperform very sparse adversarial attacks (such as one-pixel attacks), in agreement with the existing results in Figure 2.

---

### Meta-Review · Area_Chair_hTdb · 2024-12-21

**Metareview:**

This work develops adversarial attacks for group-sparse perturbations that align with semantic structures in the input. The algorithm developed is a two-part heuristic: first, the optimization via PGD of a $\ell_p$-pseudonorm (with $p=0.5$) by means of a proximal operator and, second, an optimization over the selected pixels with Nesterov accelerated gradient descent. The experiments and evaluation verify the claims and objectives of the paper.

**Strengths**

* The obtained attacks are sparser and more aligned with the semantic content of the regions that appear important, as reflected by CAM.
* Faster than other methods.
* The devise method provides an effective solution for the problem.

**Weaknesses**
* It is still not completely clear why the authors deem such attacks "interpretable", or why such a qualifier is needed.

**Summary**

In all, and while the numerical reviews reflect a borderline paper, I believe the strengths outweigh the weaknesses, and I'm recommending acceptance.

**Additional Comments On Reviewer Discussion:**

The discussion with the reviewers was productive, leading to observations and comments that have improved the paper.

**Z1qQ** had questions on the motivations behind the choice of $p=0.5$ as a regularizer, questions on the experimental settings and metrics. All of these comments were addressed.

**8hSx** had questions on the evaluation metrics, and raised careful points regarding the need for 'explainable' adversarial attacks. These were addressed by the reviewers, although this A.C. still has questions on the specific motivation for this need.

**PeSc** had questions on hyper-parameter sensitivity, experimental details and ablation studies, all of which were addressed.

**BmUV** had questions on the choice of hyperparameters and the corresponding tradeoffs. This was addressed too.

---

### Decision · Program_Chairs · 2025-01-22

Accept (Poster)